# A negative feedback loop between TET2 and leptin in adipocyte regulates body weight

Qin Zeng[1,2,6], Jianfeng Song[1,2,6], Xiaoxiao Sun[1,2], Dandan Wang[1,2], Xiyan Liao[1,2], Yujin Ding[1,2], Wanyu Hu[1,2], Yayi Jiao[1,2], Wuqian Mai[1,2], Wufuer Aini[1,2], Fanqi Wang[1,2], Hui Zhou[1,2], Limin Xie[1,2], Ying Mei[1,2], Yuan Tang[1,2], Zhiguo Xie[1,2], Haijing Wu[3], Wei Liu[4] & Tuo Deng ®[1,2,5] ✉

Ten-eleven translocation (TET) 2 is an enzyme that catalyzes DNA demethylation to regulate gene expression by oxidizing 5-methylcytosine to 5-hydroxymethylcytosine, functioning as an essential epigenetic regulator in various biological processes. However, the regulation and function of TET2 in adipocytes during obesity are poorly understood. In this study, we demonstrate that leptin, a key adipokine in mammalian energy homeostasis regulation, suppresses adipocyte TET2 levels via JAK2-STAT3 signaling. Adipocyte *Tet2* deficiency protects against high-fat diet-induced weight gain by reducing leptin levels and further improving leptin sensitivity in obese male mice. By interacting with C/EBPα, adipocyte TET2 increases the hydroxymethylcytosine levels of the *leptin* gene promoter, thereby promoting *leptin* gene expression. A decrease in adipose TET2 is associated with obesity-related hyperleptinemia in humans. Inhibition of TET2 suppresses the production of leptin in mature human adipocytes. Our findings support the existence of a negative feedback loop between TET2 and leptin in adipocytes and reveal a compensatory mechanism for the body to counteract the metabolic dysfunction caused by obesity.

The global prevalence of obesity continues to rise[1]. This trend is alarming due to the various complications of obesity, such as type 2 diabetes mellitus, cardiovascular diseases, and various malignancies, which are accompanied by a shorter life expectancy and immense economic burdens[2]. Although weight gain can be managed through lifestyle changes, surgical interventions, and a few medications, there is an unmet need to promote and sustain significant weight loss in overweight and obese individuals[3]. Consequently, a better comprehension of the mechanisms underlying weight gain could aid in the

development of therapeutic options for obesity and its associated complications.

Epigenetics is one of the key mechanisms correlating environmental factors to obesity[4]. DNA methylation is the most extensively studied epigenetic modification. The level of DNA methylation is regulated by DNA methylation and active DNA demethylation. DNA methyltransferases (DNMTs) catalyze DNA methylation by transferring a methyl to a 5' cytosine in the context of CpG to produce 5-methylcytosine (5-mC)[5]. In contrast, active DNA demethylation

[1]National Clinical Research Center for Metabolic Diseases, and Department of Metabolism and Endocrinology, The Second Xiangya Hospital of Central South University, Changsha, Hunan 410011, China. [2]Key Laboratory of Diabetes Immunology, Ministry of Education, and Metabolic Syndrome Research Center, The Second Xiangya Hospital of Central South University, Changsha, Hunan 410011, China. [3]Department of Dermatology, Hunan Key Laboratory of Medical Epigenomics, The Second Xiangya Hospital of Central South University, Changsha, Hunan 410011, China. [4]Department of Biliopancreatic Surgery and Bariatric Surgery, The Second Xiangya Hospital of Central South University, Changsha, Hunan 410011, China. [5]Clinical Immunology Center, The Second Xiangya Hospital of Central South University, Changsha, Hunan 410011, China. [6]These authors contributed equally: Qin Zeng, Jianfeng Song. ✉e-mail: dengtuo@csu.edu.cn

involves the conversion of 5-mC to 5-hydroxymethylcytosine (5-hmC) by ten-eleven translocation (TET) enzymes[6,7]. DNA methylation and active DNA demethylation are both implicated in various biological and pathological processes. Epigenetic association studies in obese patients have shown that DNA-methylated CpG sites in leukocytes are associated with body mass index (BMI) and waist circumference[4,8–11]. DNA methylation and gene expression of 2,825 genes in adipose tissue were also associated with BMI[12]. In addition, weight loss alters the DNA methylation of multiple genes in adipose tissue[13,14]. These studies demonstrate a significant association between adipose tissue DNA methylation levels and obesity. However, it remains uncertain whether the alteration of DNA methylation in adipose tissue contributes to obesity.

Recent research suggests that TETs play a governing function in energy homeostasis. By increasing β3-AR expression, loss of adipose TET proteins may protect against high-fat diet (HFD)-induced obesity[15], highlighting a pivotal regulatory role of the TET family in obesity. TET family has three members, including TET1, TET2 and TET3[16]. They all expressed in 3T3-L1 preadipocytes and involved in regulation of adipogenesis[17–20]. Intriguingly, TET2 lacks the zinc finger structure-CXXC domain, making it structurally distinct from TET1 and TET3[16]. Thus, TET2 frequently binds to gene promoters indirectly by interacting with transcription factors that regulate gene expression[21–23]. Targeting key enzymes of epigenetic regulation typically impacts the expression of a large number of genes and is accompanied by severe side effects. However, disrupting the binding of TET2 to transcription factor is safer and more specific, making TET2 a desirable therapeutic target.

Emerging evidence indicates that TET2 has a metabolic role in adipocytes. It enhances the expression of peroxisome proliferator-activated receptor γ (PPARγ) by demethylating its promoter DNA, thereby promoting adipogenesis[19]. Moreover, TET2 augments the insulin-sensitizing effect of rosiglitazone by interacting physically with PPARγ to maintain its binding to target loci[24]. It upregulates lipolysis by inducing DNA demethylation of the *Adrb3* gene promoter to increase its transcription[18]. These studies have relied on in vitro approaches, and a distinct mechanism correlating TET2-mediated epigenetic changes in adipocytes to obesity has not been established. The regulation and function of TET2 and its target genes in adipocytes remain to be investigated.

Leptin is an adipokine that signals to reduce food intake and increase energy expenditure[25]. Although leptin administration reverses obesity in congenital leptin-deficient animals and humans[26–28], it does not affect diet-induced obesity[29–31]. This phenomenon is caused by the decreased hypothalamus responsiveness to leptin in obese patients, namely "obesity-induced leptin resistance"[32,33]. Leptin resistance is a cause and a consequence of obesity[34–36]. The underlying mechanisms of leptin resistance are unknown but may involve a decrease in leptin transport across the blood-brain barrier, an upregulation of negative regulatory factors in the leptin signaling pathway, hypothalamic inflammation, endoplasmic reticulum stress, and an increase in the decomposition of leptin receptors by matrix metalloproteinase-2[37,38]. Recent research has identified hyperleptinemia as a propelling force for leptin resistance[39], which contributes to HFD-induced obesity. The partial reduction of circulating leptin levels through genetic and antibody-blocking approaches improves leptin resistance, thereby decreasing food intake and increasing energy expenditure in obese mice[40,41], providing insights into the treatment of obesity.

In this study, we demonstrate that obesity leads to a decrease in 5-hmC and TET2 levels in adipocytes, which was primarily attributed to hyperleptinemia. *Tet2* loss in adipocytes decreased leptin production, thereby improving HFD-induced leptin resistance and weight gain. TET2 directly interacted with C/EBPα to increase hydroxymethylcytosine levels and *leptin* mRNA expression by targeting the *leptin* gene promoter. Our study proposes a concept of a negative regulatory loop between TET2 and leptin in adipocytes and elucidates its function in regulating body weight.

## Results

### Obesity decreases DNA hydroxymethylation and TET2 levels in adipocytes

A genomic DNA dot blot assay of 5-hmC and 5-mC was performed to determine DNA hydroxymethylation and methylation levels in both subcutaneous adipose tissue (inguinal white adipose tissue, iWAT) and visceral adipose tissue (epididymal white adipose tissue, eWAT) from the normal diet (ND)-fed lean mice and HFD-fed obese mice. Both iWAT and eWAT from obese mice showed a significant reduction in 5-hmC levels compared with lean controls (Fig. S1a), while 5-mC levels remained unchanged (Fig. S1b). Given that the TET family of dioxygenases catalyzes the conversion of 5-mC to 5-hmC[16], we examined the gene expression levels of all three members of the TET family, including *Tet1*, *Tet2*, and *Tet3*, in adipose tissues by qRT-PCR. The mRNA levels of *Tet1* and *Tet2* were markedly decreased in both iWAT and eWAT of HFD-fed mice when compared with ND-fed mice, while the mRNA levels of *Tet3* remained unchanged (Fig. S1c). Published bulk RNA-seq data (GSE132706) revealed that the mRNA levels of *Tet2* were much higher than that of *Tet1* in adipose tissue (Fig. S1d). Moreover, western blots assay and immunohistochemical staining demonstrated that the protein levels of TET2 in iWAT and eWAT were reduced in obese mice (Fig. S1e and f). These results indicate that TET2 is the key DNA methylation eraser downregulated in WAT with obesity.

Adipocytes are the primary functional cells in adipose tissue that comprise the largest component of the tissue volume in both iWAT and eWAT. We isolated adipocytes from iWAT and eWAT and found that the levels of 5-hmC were significantly lower and the levels of 5-mC were unaltered in adipocytes from obese mice compared with lean controls (Fig. 1a, b). TET2 mRNA and protein levels were consistently and substantially decreased in adipocytes during obesity (Fig. 1c, d). Notably, the levels of *Tet2* in adipocytes from iWAT began to diminish after 4 weeks of HFD feeding, whereas it decreased after 12 weeks of HFD feeding in adipocytes from eWAT (Fig. 1e). However, there is no difference in TET2 levels between lean and obese mice in stromal vascular fraction (SVF) (Fig. S1g and h). Therefore, 5-hmC and TET2 levels in obese adipose tissue are predominantly downregulated in adipocytes but not in SVF.

### Leptin inhibits adipocyte Tet2 expression via JAK2-STAT3 signaling pathway in adipocytes

To investigate the transcriptional regulation of the adipocyte *Tet2* gene, primary differentiated adipocytes were treated with various factors known to increase in WAT with obesity. Among these factors, only leptin inhibited *Tet2* expression (Fig. 2a). A similar result was found in primary mature adipocytes (Fig. S2a). The inhibitory effect of leptin on *Tet2* mRNA expression appeared to be time-dependent (Fig. 2b) and dose-dependent (Fig. 2c) manners. The concentration of leptin in the conditioned medium (CM) collected from iWAT and eWAT of HFD-fed mice was ~300 ng/mL (Fig. 2d), exceeding the concentration required for suppression of *Tet2* expression in vitro (Fig. 2c). HFD-CM suppressed *Tet2* expression in mature adipocytes, and this suppressive effect was blocked by a leptin-neutralizing antibody (Fig. 2e). Thus, leptin is a critical negative regulator of adipocyte TET2 in vitro. To determine the regulation of TET2 by leptin in vivo, we examined *Tet2* and *leptin* mRNA levels in adipocytes from both iWAT and eWAT of HFD-fed mice and lean control mice. Our findings showed a sustained reduction in *Tet2* expression levels accompanied by a significant increase in *leptin* expression levels in adipocytes of obese WATs (Fig. S2b and c). In addition, despite clear obesity (leptin-deficient (*ob/ob*) 50 g versus control 25 g), adipocytes from *ob/ob* mice

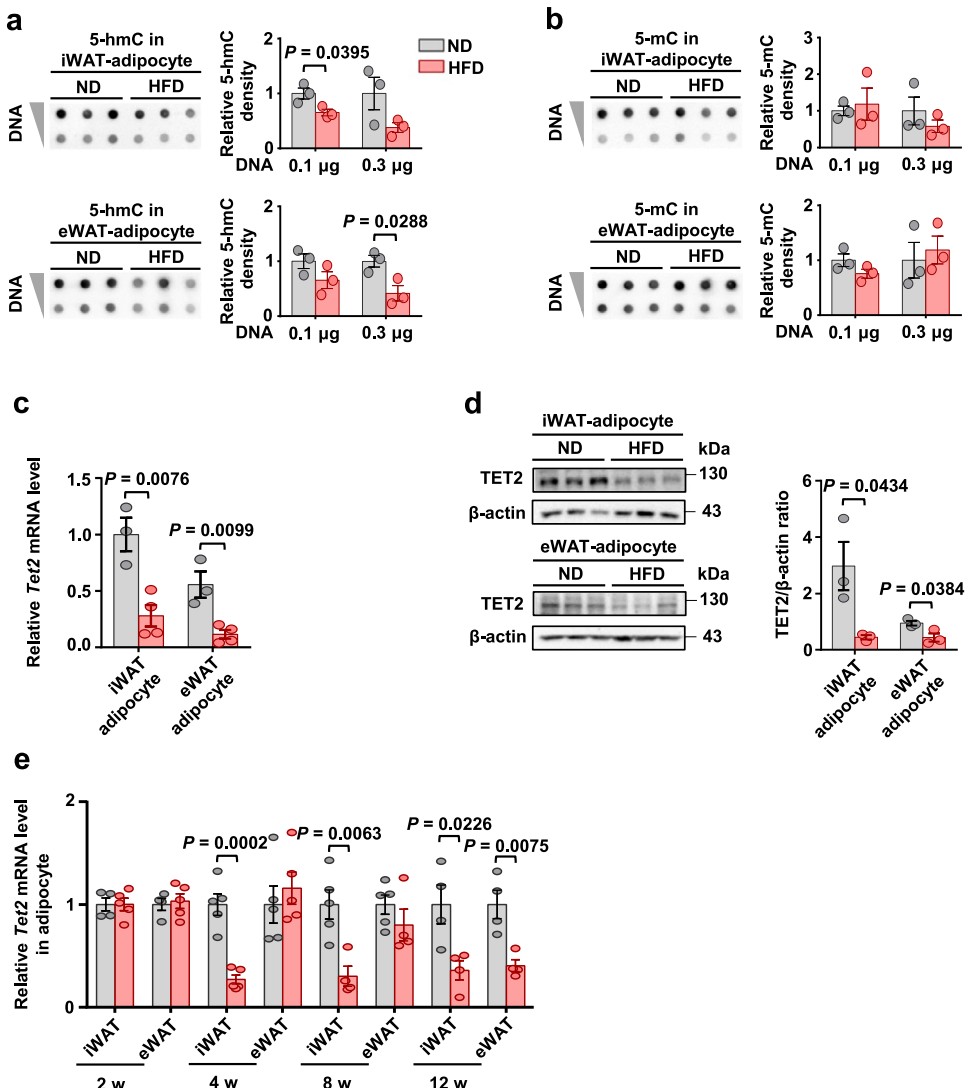

**Fig. 1 | Obesity decreases DNA hydroxymethylation and TET2 levels in adipocytes. a-b** Genomic 5-hmC (**a**) and 5-mC (**b**) levels in adipocytes of iWAT and eWAT from the C57BL/6J male mice fed either ND or HFD for 12 weeks ($n = 3$ mice/group). **c** *Tet2* mRNA levels relative to *36b4* in adipocytes of iWAT and eWAT from the mice shown in a ($n = 3$ ND, 1 ND sample was obtained by pooling samples from two mice; $n = 4$ HFD). **d** Representative immunoblot images of TET2 in adipocytes of iWAT and eWAT from the mice shown in a and densitometry analysis. *β*-actin was used as a loading control ($n = 3$ mice/group). **e** *Tet2* mRNA levels relative to *36b4* in

adipocytes of iWAT and eWAT from the C57BL/6J male mice fed either ND or HFD for 2, 4, 8, and 12 weeks (HFD 2 weeks: $n = 4$ ND and $n = 5$ HFD; HFD 4 weeks: $n = 5$ ND and $n = 5$ HFD; HFD 8 weeks: $n = 5$ ND and $n = 4$ HFD; HFD 12 weeks: $n = 4$ ND and $n = 4$ HFD; 1 ND sample was obtained by pooling samples from two mice). All data are presented as mean ± SEM. *P*-values are indicated on the graph. Statistical values are determined by two-sided unpaired Student's *t*-test. Source data are provided as a Source Data File.

exhibited higher gene and protein levels of TET2 than wild-type (WT) mice (Fig. 2f, g). Overall, obesity-induced downregulation of adipocyte TET2 is primarily attributed to hyperleptinemia.

It has been reported that the JAK2-STAT3 signaling pathway mediates multiple functions of leptin in adipocytes[42]. To investigate the specific role of the JAK2-STAT3 signaling pathway in the down-regulation of the *Tet2* gene by leptin, we examined the effects of leptin on *Tet2* gene expression in adipocytes treated with JAK2 inhibitor AZD1480, JAK2-siRNA, or STAT3-siRNA. We found that leptin-induced downregulation of *Tet2* was completely blocked by the JAK2 inhibitor AZD1480 (Fig. 2h). In addition, transfection of a JAK2-siRNA, but not a control-siRNA, reduced *Jak2* transcript levels (Fig. S2d) and rescued the inhibitory effects of leptin on *Tet2* gene expression (Fig. 2h). Moreover, the knockdown of *Stat3* by siRNA (Fig. S2e) also reversed the inhibitory effects of leptin on *Tet2* gene expression (Fig. 2i). These results indicate that leptin inhibits *Tet2* gene expression in adipocytes through the JAK2-STAT3 signaling pathway.

### Tet2 deficiency attenuates HFD-induced obesity and insulin resistance

To determine the role of TET2 in metabolic responses to caloric excess, *Tet2* deficient (*Tet2*[−/−]) and *Tet2*[+/+] mice were fed with a chow diet or HFD for 12 weeks and analyzed for metabolic phenotypes. No differences in body weight, adipocyte size, insulin sensitivity, and glucose tolerance were observed between ND-fed *Tet2*[+/+] and *Tet2*[−/−] mice (Fig. S3a–e), indicating that TET2 did not affect systemic metabolism in lean mice. At 12 weeks after HFD feeding, *Tet2*[−/−] mice have ~20% lower body weight than *Tet2*[+/+] mice (Figs. 3a and S4a). A body composition analysis showed that the weight differences between *Tet2*[−/−] and *Tet2*[+/+] mice were due to a decrease in fat mass (Fig. 3b). Notably, adipose tissue and liver weights were decreased in *Tet2*[−/−] mice (Figs. 3c and S4b). In addition, the adipocyte size was smaller (Fig. S4d), and lipid accumulation in brown adipose tissue (BAT) and liver was markedly reduced (Figs. S4c and e) in *Tet2*[−/−] mice compared with control littermates. Consistent with reduced adiposity, *Tet2*[−/−]

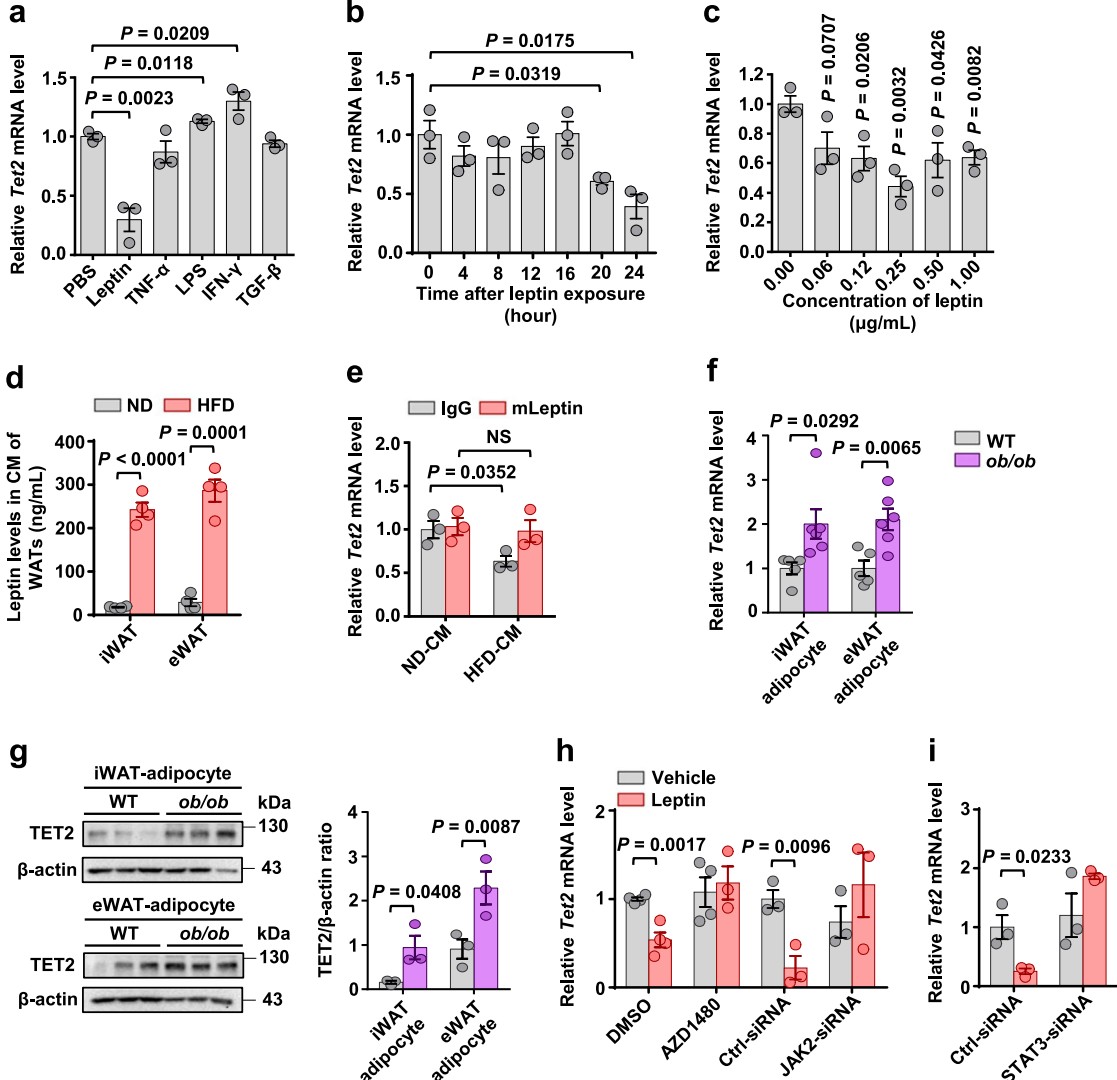

**Fig. 2 | Leptin inhibits adipocyte Tet2 expression via JAK2-STAT3 signaling pathway in adipocytes. a** *Tet2* mRNA levels relative to *36b4* in differentiated mature adipocytes that were treated with PBS, leptin (1 μg/mL), TNF-α (5 ng/mL), LPS (500 ng/mL), IFN-γ (2 ng/mL), or TGF-β1 (10 ng/mL) for 24 h (*n* = 3). **b-c** *Tet2* expression in differentiated mature adipocytes treated with leptin at various time points (**b**) and concentrations (**c**). Time points and concentrations are indicated on the graphs (*n* = 3). **d** Leptin levels in conditioned medium (CM) from iWAT and eWAT of ND-fed and HFD-fed mice (*n* = 4). **e** *Tet2* expression in mature adipocytes treated with control antibody (IgG) or leptin neutralizing antibody (mLeptin) for 1 h, followed by treating with CM from eWAT of ND-fed and HFD-fed mice for another 24 h (*n* = 3). **f** *Tet2* mRNA levels relative to *36b4* in adipocytes of iWAT and eWAT from WT and *ob/ob* mice (*n* = 5 WT, *n* = 6 *ob/ob*). **g** Representative immunoblot images of TET2 in adipocytes of iWAT and eWAT from the mice shown in **f** and densitometry analysis. β-actin was used as a loading control (*n* = 3 mice/group). **h** *Tet2* expression in differentiated 3T3-L1 adipocytes treated with DMSO (*n* = 4) or AZD1480 (JAK2 inhibitor, *n* = 3), Ctrl-siRNA (*n* = 3) or JAK2-siRNA (*n* = 3) for 24 h, followed by treating with leptin for another 24 h. **i** *Tet2* expression in differentiated 3T3-L1 adipocytes treated with Ctrl-siRNA or STAT3-siRNA for 24 h, followed by treating with leptin for another 24 h (*n* = 3). All data are presented as mean ± SEM. *n* indicates the number of biologically independent samples examined. *P*-values are indicated on the graph. Statistical values are determined by two-sided unpaired Student's *t*-test. Source data are provided as a Source Data File.

mice fed with HFD had greater insulin sensitivity (Fig. 3d, e), lower fasting plasma glucose (FPG) (Fig. 3f), and greater glucose tolerance (Fig. 3g). To elucidate the possible mechanism of the reduced body weight in HFD-fed *Tet2⁻/⁻* mice, we investigated the energy expenditure and food intake in *Tet2⁻/⁻* and *Tet2⁺/⁺* mice at 5 weeks of HFD before a significant difference in body weight was observed. HFD-fed *Tet2⁻/⁻* mice had higher oxygen consumption (Fig. 3h, i) and energy expenditure (Fig. 3j), and less food intake (Fig. 3k) than *Tet2⁺/⁺* mice. No difference was observed in physical activity between these two groups of mice (Fig. S4f). Consistent with the alterations in energy expenditure, *Tet2⁻/⁻* mice fed HFD exhibited higher mRNA levels of thermogenic genes in BAT (e.g., *Ucp1* and *Cidea*), iWAT (e.g., *Prdm16*, *Ppargc1a* and *Cidea*) and eWAT (e.g., *Ucp1*, *Prdm16* and *Ppargc1a*) compared with *Tet2⁺/⁺* mice (Fig. 3l–n). These results suggest that

deficiency of *Tet2* leads to an increase in BAT activity and WAT browning during obesity, ultimately resulting in elevated energy consumption and reduced weight gain. Additionally, we obtained adipose-derived stem cells (ASCs) from *Tet2⁺/⁺* and *Tet2⁻/⁻* mice to perform in vitro and in vivo adipogenesis experiments, confirming that *Tet2* deficiency effectively reduced adipogenesis (Fig. S4g–j). To exclude the possibility that TET1 and TET3 compensate for some dioxygenase activities under conditions of *Tet2* loss in adipose tissue, we examined the mRNA levels of *Tet1* and *Tet3* in iWAT and eWAT from lean *Tet2⁻/⁻* and *Tet2⁺/⁺* mice and found that their expression levels did not differ between the two groups (Fig. S4k). Collectively, these results indicate that *Tet2* deficiency ameliorates HFD-induced obesity and insulin resistance by promoting energy consumption and inhibiting food intake.

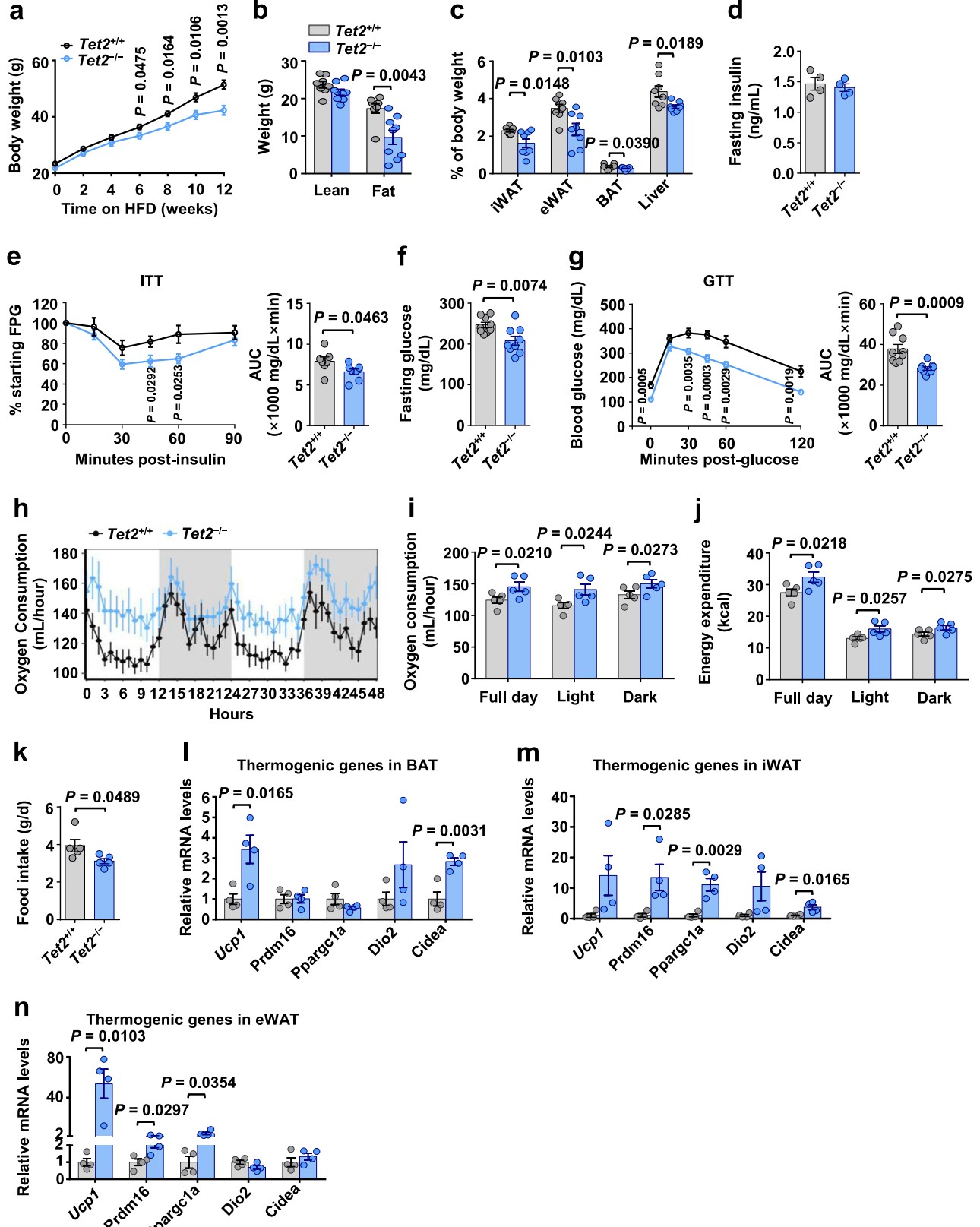

## Adipocyte-specific Tet2 deficiency protects against HFD-induced obesity and insulin resistance

To define the role of TET2 in adipocytes in regulating obesity and insulin resistance, we generated adipocyte-specific *Tet2* knockout mice (AKO) by crossing *Tet2^fl/fl* mice to *Adipoq*-Cre transgenic mice (Fig. S5a). AKO mice were viable and born at the expected Mendelian ratio, and

there were no discernible morphological differences between the two genotypes. As expected, *Tet2* expression was decreased in all the examined fat depots of ND-fed AKO mice compared with the AWT (*Tet2^fl/fl*) group, whereas no changes were shown in non-adipose tissues (Fig. S5b). *Tet2* expression was similarly reduced in adipocytes but not SVF (Fig. S5c). These results indicated that the construction of the AKO

**Fig. 3 | Tet2 deficiency attenuates HFD-induced obesity and insulin resistance.** **a** and **b** Body weight progression (**a**) and body fat (**b**) in *Tet2*^+/+^ and *Tet2*^−/−^ mice fed HFD for 12 weeks, starting at 6 weeks of age ($n = 8$ *Tet2*^+/+^; $n = 9$ *Tet2*^−/−^). **c** Relative tissue weights of iWAT, eWAT, BAT, and liver mass after 12 weeks of HFD feeding ($n = 8$ *Tet2*^+/+^; $n = 8$ *Tet2*^−/−^). (**d**) Fasting insulin levels after 12 weeks of HFD feeding ($n = 4$ mice/group). **e** ITT and its respective area under the curve (AUC) after 12 weeks of HFD feeding ($n = 7$ *Tet2*^+/+^; $n = 8$ *Tet2*^−/−^). **f** Fasting glucose levels after 12 weeks of HFD feeding ($n = 9$ mice/group). **g** GTT and its respective AUC after 12 weeks of HFD feeding ($n = 9$ mice/group). **h–k** Changes in oxygen consumption (**h**) at different time points, oxygen consumption (**i**) and energy expenditure (**j**) during light, dark hours, and full day, and daily food intake (**k**) after 5 weeks of HFD feeding ($n = 5$ mice/group). **l–n** mRNA levels of thermogenic genes relative to *β-actin* in BAT (**l**), iWAT (**m**) and eWAT (**n**) after 5 weeks of HFD feeding ($n = 4$ mice/group). All data are presented as mean ± SEM. *P*-values are indicated on the graph. Statistical values are determined by two-sided unpaired Student's *t*-test in (**a–g**) and (**k–n**), one-way ANCOVA test in (**i**) and (**j**). Source data are provided as a Source Data File.

mouse model was successful. AWT and AKO littermate mice were subjected to ND or HFD for 12 weeks and analyzed for differences in metabolic phenotypes. Under ND feeding conditions, AKO mice exhibited comparable body weight, adipocyte size, insulin sensitivity, and glucose tolerance with AWT controls (Figs. S5d–h). However, after 12 weeks of HFD feeding, AKO mice gained less body weight and adipose mass than AWT mice (Figs. 4a, b, and S6a). In addition, reduced eWAT weight and adipocyte size (Fig. 4c and S6b–d) and decreased hepatic lipid accumulation (Fig. S6e) were accompanied by improved insulin sensitivity and glucose tolerance (Fig. 4d–g) in AKO mice on HFD. Calorimetry analysis by CLAMS revealed that AKO mice exhibited higher total oxygen consumption (Fig. 4h, i), energy expenditure (Fig. 4j), and less food intake (Fig. 4k) than AWT mice at 5 weeks of HFD. Consistently, HFD-fed AKO mice displayed higher mRNA levels of thermogenic genes, such as *Prdm16, Ppargc1a* and *Cidea* in BAT and *Ucp1, Ppargc1a* and *Dio2* in eWAT, when compared with WT controls (Fig. 4l–n). Additionally, treatment of ASCs from AKO mice with adipocyte differentiation induction cocktail resulted in no difference in adipogenic capacity compared with ASCs from AWT mice (Figs. S6f and g), helping to exclude the impact of adipogenesis on the reduction of adiposity observed in AKO mice during obesity. Overall, these results show that conditional *Tet2* deletion in adipocytes protects against HFD-induced obesity and insulin resistance, mainly ascribed to an increase in energy consumption and a reduction in food intake.

## Tet2 deficiency improves HFD-induced obesity and insulin resistance by partially reducing leptin levels

As leptin is a classic adipokine known to reduce food intake and increase energy expenditure, we hypothesized that *Tet2* deficiency in adipocytes may increase plasma leptin levels. Surprisingly, under both ND- and HFD-feeding conditions, plasma leptin levels in AKO mice were significantly lower than in control mice (Fig. 5a). Reduction of leptin levels in obese rodents has been shown to effectively increase leptin sensitivity and cause weight loss[40,43], implying that reducing plasma leptin levels may increase leptin sensitivity in obese AKO mice. Indeed, HFD-fed AKO mice exhibited a greater reduction of food intake at 24 h and 30 h after acute injection of leptin (Fig. 5b). Additionally, AKO mice had higher p-STAT3 levels in the arcuate nucleus of the hypothalamus than their WT counterparts under both saline and leptin injection conditions (Fig. 5c), indicating that *Tet2*-deficient mice are more sensitive to leptin.

Next, we designed two experiments to clarify the role of leptin in the regulation of body weight by adipocyte TET2. In experiment 1, *Tet2* and *leptin* double knockout (*Tet2*^−/−^ *ob/ob*) and control (*Tet2*^+/+^ *ob/ob*) mice were fed HFD for 14 weeks to examine the metabolic effects of *Tet2* deficiency in the context of *leptin* deficiency. There were no differences in body weight (Fig. 5d and S7a), tissue size (Fig. S7b), total oxygen consumption (Figs. S7c and d), energy expenditure (Fig. S7e), and food intake (Fig. S7f) between *Tet2*^+/+^ *ob/ob* and *Tet2*^−/−^ *ob/ob* mice after HFD challenge. Consistently, insulin sensitivity (Fig. 5e) and glucose tolerance (Fig. 5f) did not differ between the two groups. In experiment 2, AWT and AKO mice were fed HFD, and AKO mice were supplemented with leptin by long-term intraperitoneal injection to reach plasma leptin levels similar to PBS-injected AWT mice (Fig. 5g).

Leptin supplementation effectively eliminated the differences in body weight (Fig. 5h), tissue weight (Figs. S8a and b), insulin sensitivity (Fig. 5i), and glucose tolerance (Fig. 5j) between the two groups. In addition, AKO mice supplemented with leptin displayed no significant difference in oxygen consumption (Fig. 5k, l) and food intake (Fig. 5m) compared with PBS-injected AWT mice. Furthermore, we examined adaptive thermogenesis in *Tet2*-deficient and double knockout (*Tet2*^−/−^ *ob/ob*) mice. The cold-induced thermogenesis response in *Tet2* deficient mice was enhanced compared with that in the control groups after 5 weeks of HFD feeding (Figs. S7g–j). However, no such difference was observed between double knockout mice and control mice (Figs. S7k–n). Together, these results suggest that either *leptin* deficiency or leptin supplement normalized the metabolic changes induced by *Tet2* deficiency.

## TET2 upregulates leptin gene expression via interacting with C/EBPα in adipocytes

To gain insight into whether *leptin* gene expression is regulated by TET2 in adipocytes, we examined *leptin* mRNA levels in adipocytes from AKO and AWT mice. Under both lean and obese conditions, *leptin* mRNA levels in adipocytes were decreased in AKO mice when compared with AWT mice (Fig. 6a). Additionally, the knockdown of *Tet2* significantly downregulated *leptin* gene expression in differentiated primary adipocytes (Fig. 6b), confirming that TET2 controls *leptin* gene expression in adipocytes. To identify *leptin* as a target gene of TET2, chromatin immunoprecipitation sequencing (ChIP-seq) was performed on TET2-enriched chromatin harvested from differentiated primary adipocytes. As shown in Fig. S9a, ChIP-seq signal in a 6 kb region flanking significant TET2 peak near the transcriptional start site (TSS). TET2 peaks (14103 peaks) were distributed across the genome, with the most enrichment on chromosomes 2 and 4 (Fig. S9b, Supplementary Data 1), and overrepresentation at promoter (Fig. S9c). Gene ontology analysis of biological processes revealed that TET2 antibody immunoprecipitates were enriched in genes associated with cellular process, cell communication and structure development (Fig. S9d). As anticipated, sequencing results revealed a prominent TET2 peak in the proximal promoter region of the *leptin* gene, verifying that *leptin* is a target gene of TET2 (Fig. 6c). Next, we conducted ChIP-qPCR with specific primers to verify the binding of TET2 to the promoter of the *leptin* gene. Consistent with the results of sequencing, this assay revealed an enrichment of TET2 at the promoter of the *leptin* gene (Fig. 6d). To further investigate how TET2 regulates *leptin* expression, we employed a TET2 enzymatic inhibitor, Bobcat339[44], to assess its impact on *leptin* expression in mature adipocytes. After confirming its efficacy in inhibiting 5-hmC (Fig. S9e), treatment with Bobcat339 resulted in reduced mRNA levels of *leptin* (Fig. 6e) in adipocytes. This suggests that TET2-mediated 5-hmC modification plays a regulatory role in adipocyte *leptin* expression. Furthermore, we performed hydroxymethylated DNA immunoprecipitation qPCR (hMeDIP-qPCR) and methylated DNA immunoprecipitation qPCR (MeDIP-qPCR) in adipocytes isolated from eWAT of ND-fed *Tet2*^+/+^ and *Tet2*^−/−^ mice. We observed that the 5-hmC levels were markedly decreased while 5-mC levels were increased on *leptin* gene promoter in adipocytes of *Tet2*^−/−^ mice in contrast with *Tet2*^+/+^ mice (Fig. 6f, g).

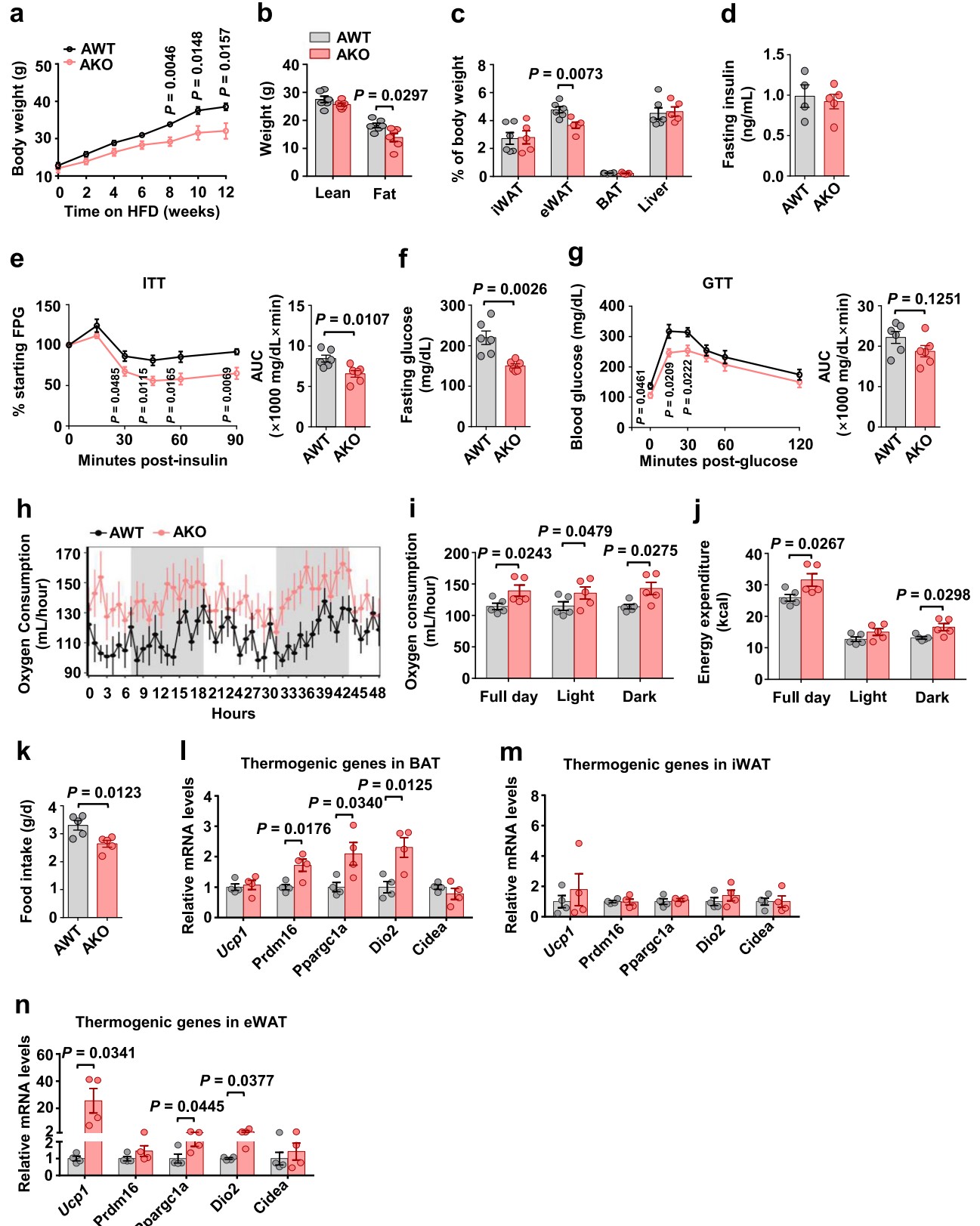

Since TET2 lacks a DNA binding domain and must interact with transcription factors to modulate the expression of its target genes[21–23], we further analyzed the TET2 binding motifs in the proximal promoter region of the *leptin* gene to identify the key transcription factor(s). ChIP-seq data revealed the presence of three binding motifs in the TET2 peak near the TSS of the *leptin* gene, namely SP1, C/EBPα, and AP-

2β (Fig. 6c). Among these, AP-2β has been reported to inhibit the expression and secretion of leptin[45], so it is improbable that it would aid TET2 in promoting leptin expression. To determine whether SP1 or C/EBPα mediates the positive regulation of *leptin* gene expression by TET2, we suppressed the expression of two other transcription factors, SP1 and C/EBPα, using siRNA. We found that C/EBPα, but not SP1,

**Fig. 4 | Adipocyte-specific Tet2 deficiency protects against HFD-induced obesity and insulin resistance. a**, **b** Body weight progression (**a**) and body fat (**b**) in AWT and AKO mice fed HFD for 12 weeks, starting at 6 weeks of age ($n = 6$ mice/group). **c** Relative tissue weights of iWAT, eWAT, BAT, and liver mass after 12 weeks of HFD feeding ($n = 6$ AWT; $n = 5$ AKO). **d** Fasting insulin levels after 12 weeks of HFD feeding ($n = 4$ AWT; $n = 5$ AKO). **e** ITT and its respective AUC after 12 weeks of HFD feeding ($n = 6$ mice/group). **f** Fasting glucose levels after 12 weeks of HFD feeding ($n = 6$ mice/group). **g** GTT and its respective AUC after 12 weeks of HFD feeding ($n = 6$ mice/group). **h**–**k** Changes in oxygen consumption (**h**) at different time points, oxygen consumption (**i**) and energy expenditure (**j**) during light, dark hours, and full day, and daily food intake (**k**) after 5 weeks of HFD feeding ($n = 5$ mice/group). **l**–**n** mRNA levels of thermogenic genes relative to $\beta$-actin in BAT (**l**), iWAT (**m**) and eWAT (**n**) after 5 weeks of HFD feeding ($n = 4$ mice/group). All data are presented as mean ± SEM. *P*-values are indicated on the graph. Statistical values are determined by two-sided unpaired Student's *t*-test in (**a**–**g**) and (**k**–**n**), one-way ANCOVA test in (**i**) and (**j**). Source data are provided as a Source Data File.

affected the enrichment of TET2 at the promoter of the *leptin* gene (Fig. 6h). Additionally, Co-IP experiments revealed a physical association between TET2 and C/EBPα in mature adipocytes (Fig. 6i). Furthermore, ChIP and sequential ChIP (reChIP) assays confirmed the co-recruitment of C/EBPα and TET2 at the *leptin* gene promoter (Fig. 6j). Knockdown of *Tet2* gene didn't influence the expression of C/EBPα in adipocytes (Fig. S9f). Similarly, knockdown of *Cebpa* gene didn't impact the expression of TET2 in adipocytes (Fig. S9g). These findings suggest that TET2 and C/EBPα don't affect each other's expression levels in adipocytes. To assess the enrichment of C/EBPα at the promoter of the *leptin* gene in adipocytes of WATs under both lean and obese conditions, we performed ChIP-qPCR and confirmed no disparity in the binding capacity of C/EBPα to the *leptin* gene promoter between ND-fed and HFD-fed mice (Fig. 6k). Collectively, our results show that TET2 binds to the *leptin* gene promoter by interacting with C/EBPα, inducing demethylation and expression of the *leptin* gene.

### TET2 levels are negatively correlated with LEPTIN levels and BMI in humans

To provide clinical support for the findings in mice, we analyzed the 5-hmC and 5-mC levels in human subcutaneous adipose tissue (SAT). In the SAT of obese individuals, 5-hmC levels were significantly reduced, whereas 5-mC levels were elevated (Fig. 7a, b). To explore the association between TET2 and leptin in human obesity, we detected the expression levels of *TET2* and *LEPTIN* in SAT from a cohort of 15 subjects classified by BMI (clinical parameters are summarized in Table S1). As shown in Fig. 7c, adipose *TET2* mRNA levels were lower, while *LEPTIN* mRNA levels were higher in obese than in nonobese subjects. In this cohort, *TET2* expression inversely correlated with *LEPTIN* levels (Fig. 7d). To gain insight into the relationship between *TET2* and *LEPTIN* gene expressions in human adipocytes, we reanalyzed our previous microarray data (GSE44000) from human adipocytes and found that *TET2* levels were negatively associated with *LEPTIN* levels (Fig. 7e) and BMI (Fig. 7f). Moreover, treatment of Bobcat339, a TET2 inhibitor, led to a decreased production of leptin in mature adipocytes (Fig. 7g). These human data indicate that obesity-related hyperleptinemia is associated with a decline in TET2 expression in adipocytes, supporting a negative feedback loop between TET2 and leptin in the context of obesity.

### Discussion

Recent research has shown that abnormal DNA methylation levels in adipocytes can be a consequence of obesity[46]. In contrast, the aberrant alteration of DNA methyltransferase or DNA demethylation enzyme may also play a role in the development of obesity[15,47,48]. However, the regulation mechanism between DNA methylation and obesity remains inadequately understood. Here, we find that obesity decreases DNA hydroxymethylation and TET2 levels in adipocytes via the leptin signaling pathway. Furthermore, we identify the crucial role of adipocyte TET2 in HFD-induced obesity and elucidate that TET2 interacts with transcription factor C/EBPα to control *leptin* gene expression, thereby regulating body weight.

According to a study on adipocytes, the expression of TET2 is reduced under HFD condition[24], which is consistent with our findings (Fig. 1c, d). However, the regulator responsible for the downregulation

of TET2 in obese adipocytes remains unknown. In our study, we demonstrated that hyperleptinemia in HFD-induced obese mice decreased TET2 levels in adipocytes (Fig. 2e). Consistently, in the leptin-deficient (*ob/ob*) obese mice, adipocyte expression of TET2 were elevated when compared with lean controls (Fig. 2f, g), suggesting that leptin, rather than obesity, negatively regulated the expression of TET2 in adipocytes. Notably, leptin levels nearly double on the first day of the HFD challenge and continue to rise with obesity[40]. However, the downregulation of the *Tet2* gene in adipocytes of obese mice occurred at about 4 weeks of HFD feeding (Fig. 1e), which was far from the time when leptin was induced, implying that leptin needs to accumulate to a high concentration to exert its inhibitory effect. Indeed, leptin inhibited the expression of the *Tet2* gene in primary differentiated adipocytes at concentrations >0.12 μg/mL (Fig. 2c). Additionally, obesity-induced decline of adipocyte *Tet2* levels is much later in eWAT than in iWAT. To explain the temporal discrepancy in the reduction of adipocyte *Tet2* levels in iWAT and eWAT during obesity, we investigated the leptin levels in these two types of adipocytes. Our results revealed that the mRNA levels of *leptin* increased in both iWAT and eWAT adipocytes at 2 weeks of HFD and continued to rise until 12 weeks of HFD (Figs. S2b and c). The secretion levels of leptin from these adipose tissues were also similar under obese conditions (Fig. 2d). However, as shown in Fig. S2a, leptin suppressed *Tet2* expression more prominently in iWAT adipocytes compared with eWAT adipocytes. This suggests that iWAT adipocytes are more sensitive to leptin-induced *Tet2* inhibition. Therefore, it appears that the reduced sensitivity to leptin, rather than leptin concentration, contributes to the delayed decline of adipocyte *Tet2* levels in eWAT during obesity. In humans, the expression levels of *TET2* and *LEPTIN* were negatively correlated in adipocytes and adipose tissue of SAT (Fig. 7d, e). Thus, leptin is an important determinant of the expression of TET2 in adipocytes.

Compelling evidence suggests that DNA methylation plays an important role in the regulation of leptin expression. In human adipocytes, the *leptin* gene promoter is hypomethylated and its expression level is elevated[49]. During adipogenesis, the promoter region of the *leptin* gene is demethylated, resulting in a growing expression level[50]. Moreover, DNMT inhibitors can promote *leptin* gene expression in mature 3T3-L1 cells[51]. These studies revealed an association between DNA methylation and *leptin* gene expression, whereas the role of DNA demethylation in the regulation of *leptin* gene expression is unknown. Using genetic and siRNA approaches to knockdown *Tet2* gene in adipocytes, we found that loss of *Tet2* effectively suppressed the mRNA levels of *leptin* in adipocytes (Fig. 6a, b), accompanied by the decrease of 5-hmC levels and increase of 5-mC levels at leptin promoter (Fig. 6f, g). These results indicate that TET2 regulates DNA demethylation at the leptin promoter, thereby promoting *leptin* gene expression. Thus, TET2 is a crucial epigenetic regulator of leptin expression.

Emerging evidence suggests that TET2 may play a regulatory function in metabolic disturbances. In vitro research demonstrated the physical interaction between TET2 and PPARγ in adipocytes, which increased the insulin-sensitizing effect of rosiglitazone[24]. Fuster et al. reported that the loss of TET2 function in hematopoietic cells could promote the secretion of IL-1β in macrophages, thereby exacerbating obesity-induced insulin resistance[52]. However, the in vivo functions of adipocyte TET2 remain unknown. Our data showed that both global

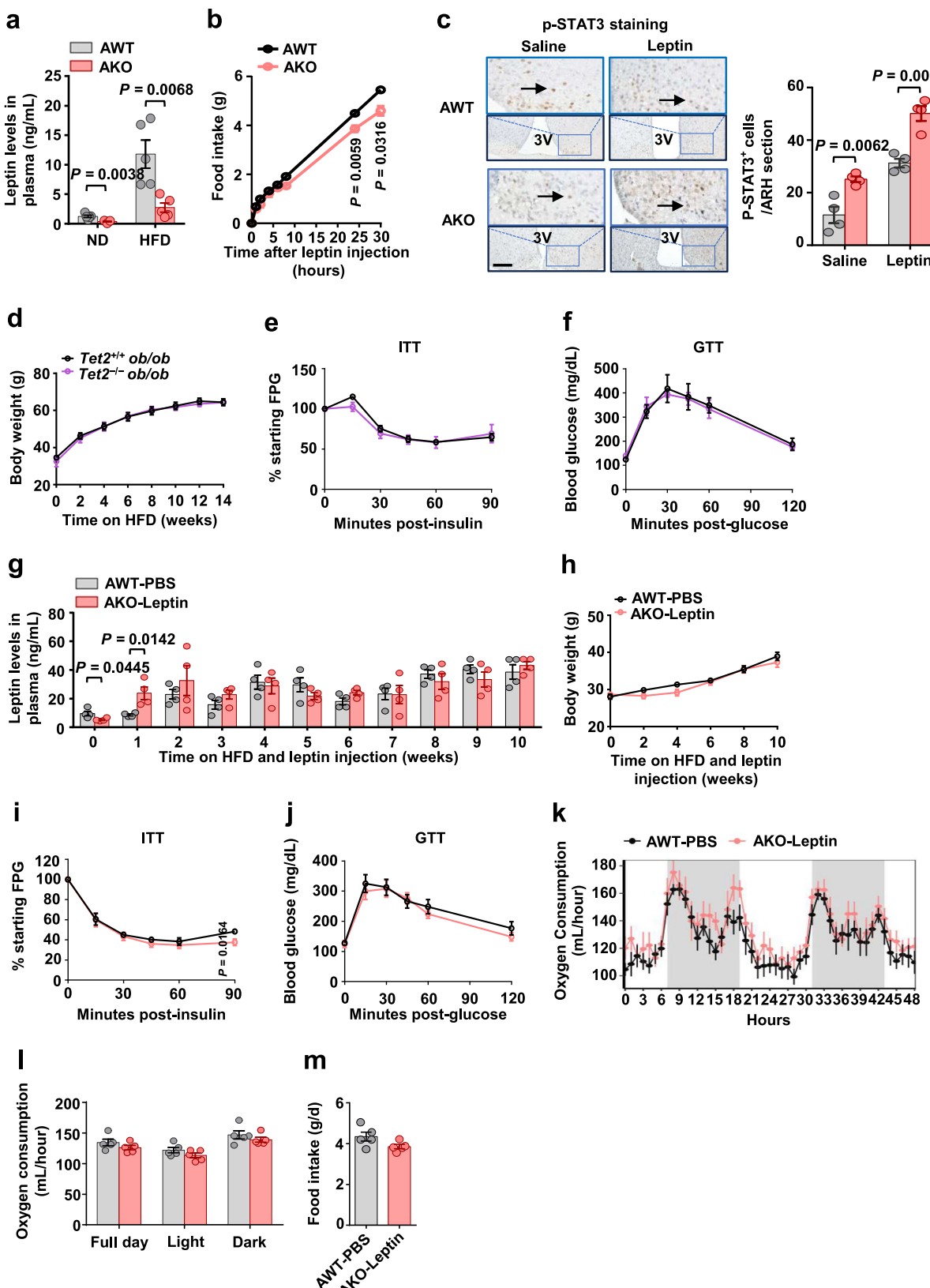

and adipocyte-specific *Tet2* deficiency protected against HFD-induced obesity in tandem with increased energy expenditure and decreased food intake (Figs. 3 and 4), indicating an increase in leptin sensitivity. Indeed, the deletion of *Tet2* in adipocytes increased the expression of p-STAT3 in the hypothalamus (Fig. 5c), which is known to be associated with enhanced leptin sensitivity. Our study illuminates a function of

adipocyte TET2 in regulating energy metabolism and raises an epigenetic mechanism involved in energy metabolism.

Overcoming obesity-induced leptin resistance has been a challenge for a long time. A previous study demonstrated that hyperleptinemia is required for the development of leptin resistance[39]. Additionally, Tam et al. found that CB1 receptor agonists reduced body

**Fig. 5 | Tet2 deficiency improves HFD-induced obesity and insulin resistance by partially reducing leptin levels. a** Leptin levels in plasma of AWT and AKO mice after 11 weeks of ND feeding or 5 weeks of HFD feeding ($n = 5$ mice/group). **b** Effects of acute leptin injections on food intake in AWT and AKO mice after overnight fasting ($n = 6$ mice/group). **c** DAB staining of p-STAT3 (3 V: Third ventricle; the arrows represent p-STAT3 positive expression, scale bars, 100 μm) after saline and leptin injection in AWT and AKO mice ($n = 4$ mice/group). **d** Body weight progression in *Tet2*[+/+] *ob/ob* and *Tet2*[−/−] *ob/ob* mice fed HFD for 14 weeks, starting at 5 weeks of age ($n = 5$ mice/group). **e** ITT from the mice shown in (**d**) ($n = 5$ mice/group). **f** GTT from the mice shown in (**d**) ($n = 5$ mice/group). **g** Leptin levels in plasma of

AWT-PBS and AKO-Leptin mice supplemented with PBS or leptin for 10 weeks, starting at 5 weeks of HFD ($n = 4$ mice/group). **h–j** Body weight progression (**h**), ITT (**i**) and GTT (**j**) in mice shown in (**g**) ($n = 7$ mice/group). **k–m** Changes in oxygen consumption (**k**) at different time points, oxygen consumption (**l**) during light, dark hours, and full day, and daily food intake (**m**) after 10 weeks of PBS or leptin supplementation ($n = 5$ mice/group). All data are presented as mean ± SEM. *P*-values are indicated on the graph. Statistical values are determined by two-sided unpaired Student's *t*-test in (**a–j**) and (**m**), one-way ANCOVA test in (**l**). Source data are provided as a Source Data File.

weight in obese mice by improving leptin sensitivity and reversing hyperleptinemia[43]. Most recently, Zhao et al. revealed that hyperleptinemia is the driving force of HFD-induced leptin resistance, and partial reduction of circulating leptin levels by gene suppression or neutralizing antibodies restored leptin sensitivity and led to weight loss[40]. Our study demonstrated that loss of *Tet2* in adipocytes led to the reduction of circulating leptin levels (Fig. 5a) and a marked increase in basal and leptin-induced p-STAT3 levels in the hypothalamus (Fig. 5c), accompanied by resistance to HFD-induced obesity (Fig. 4a). To confirm the involvement of leptin in adipocyte *Tet2* deficiency-induced weight loss, we conducted leptin deletion and supplement experiments in the context of *Tet2* deficiency and found that the elimination of the discrepancy of leptin levels between WT and *Tet2*-deficient mice led to similar metabolic phenotypes in these two groups (Fig. 5d–m). These results indicate that *Tet2* deficiency in adipocytes improves HFD-induced leptin resistance by partially reducing leptin levels and subsequently leads to weight loss, supporting that partially reducing circulating leptin levels may be a promising strategy to control body weight.

Due to the dynamic and reversible nature of epigenetic modifications, there has been considerable interest in the development of epigenetic therapies for a variety of diseases[53–55]. Nevertheless, the pleiotropic effects derived from the complexity of the epigenome limit the use of epigenetic drugs. TET2 lacks a DNA-binding domain and typically regulates the expression of target genes by interacting with transcription factors. Thus, targeting the interaction between TET2 and its partner transcription factor makes epigenetic therapies more specific. Using ChIP-seq and Co-IP, we determined that TET2 bound indirectly to the *leptin* gene promoter by binding to C/EBPα (Fig. 6c, i). Our study reveals a mechanism for the regulation of *leptin* gene expression in adipocytes, identifying the TET2-C/EBPα interaction as a specific epigenetic target for controlling leptin expression. A recent study found that C/EBPα interacted with TET2 and recruited it to specific regions of DNA, maintaining high levels of 5-hmC in the gene promoter region of pluripotency genes during B cell reprogramming[23]. As a result, disruption of the interaction between TET2 and C/EBPα should be approached with caution, as it may influence the development of B cells.

In conclusion, our study reveals a feedback loop between TET2 and leptin (Fig. 7h), which enables adipocytes to coordinate the hyperleptinemia associated with obesity. In this negative feedback loop, leptin plays a predominant role, as leptin inhibits adipocyte TET2 significantly in the context of obesity. The hyperleptinemia-induced reduction of adipocyte TET2 appears to be a compensatory response aimed at suppressing the elevated levels of leptin. Although the decrease in TET2 expression in adipocytes did not prevent the increase in leptin levels during obesity, the loss of *Tet2* in obese mice resulted in the downregulation of leptin levels and an enhancement of leptin sensitivity, which ultimately led to weight loss. This suggests that the downregulation of TET2 in adipocytes can promote weight loss by reducing leptin levels. Our mechanistic studies reveal that TET2 regulates the expression of the *leptin* gene by interacting with C/EBPα and increasing the levels of hydroxymethylcytosine at the leptin promoter. Therefore, there is a possibility of developing a

therapeutic strategy for weight loss by inhibiting the interaction between TET2 and C/EBPα in adipocytes.

## Methods
### Human samples
This study was conducted in accordance with the Declaration of Helsinki and was approved by the Ethics Committee of the Second Xiangya Hospital of Central South University (No. LYF2022207). Written informed consent was obtained from all human donors prior to their enrollment in the study. Human SAT samples were collected from two groups: obese donors (BMI ≥ 30 kg/m²) who met the recruitment criteria for bariatric surgery, and nonobese donors (BMI < 30 kg/m²) undergoing non-acute cholecystectomy surgery. SAT consisted of 100 mg tissue blocks. Dissociation of the SAT was immediately processed (within 2 h of removal from the donors), followed by DNA and RNA extraction.

### Animals
All animal studies were performed in accordance with procedures approved by the Central South University Animal Care and Use Committee. All mice were housed in a specific pathogen-free animal facility at 22 °C with a 12 h light/dark cycle and 50–60% relative humidity. Mice had free access to water and food, either an ND (1010001, Jiangsu Xietong Pharmaceutical Bio-engineering Co.LTD) or an 60% HFD (D12492, Wuhan BIOPIKE Bioscience Co.LTD). 6 week-old C57BL/6J mice and 12 week-old C57BL/6J *ob/ob* (*Lep*[ob]/*Lep*[ob]) mice were purchased from Slac Laboratory Animal Inc and the Model Animal Research Center of Nanjing University, respectively. Heterozygous *leptin* knockout mouse (*ob*/+) mice (Stock: 19006 A) were purchased from Jiangsu Wukong Biotechnology Co. LTD. *Tet2*[−/−] mice were generated and kindly provided by Dr. Xu's group[56]. *Adipoq*-Cre mice (B6.FVB-Tg (*Adipoq*-Cre)1Evdr/J; Stock 028020) and *Tet2*[fl/fl] mice (B6;129S-*Tet2*[tm1.1Iaai]/J; Stock 017573) were both obtained from the Jackson Laboratory and bred to generate experimental mice groups, including *Tet2*[fl/fl] (AWT) and *Tet2*[AdipoqCre](AKO). *Tet2*[−/−] mice were crossed with *ob*/+ mice to generate *Tet2* and *leptin* double knockout (*Tet2*[+/+] *ob/ob*) mice. The sex of mice used for experiments is male, as male mice exhibited a significantly higher percentage of visceral fat, body fat[57] and weight gain[58], worsened glucose tolerance[58] and decreased insulin sensitivity[59] compared with female mice. 5–6 week-old male mice were fed with HFD to establish the diet-induced obesity model. These male mice were euthanized via inhalation of carbon dioxide ($CO_2$) to induce asphyxiation, followed by cervical dislocation as a secondary euthanasia, before tissue isolation.

### 3T3-L1 cell line
Murine embryonic 3T3-L1 cells (ATCC CL-173, Manassas, VA) were cultured in DMEM-High Glucose medium supplemented with 10% FBS, 100 IU/mL of penicillin, and 100 mg/mL of streptomycin solution, at 5% $CO_2$ and 37 °C in a humidified cell incubator. To induce adipocyte differentiation, confluent 3T3-L1 cells were incubated with an adipogenic cocktail, including 0.5 mM isobutylmethylxanthine (IBMX, Sigma-Aldrich, Cat#I5879-1 G), 1 μM dexamethasone (Sigma-Aldrich, Cat#D4902) and 10 μg/mL insulin (Gibco, Cat#12585014). After two days, the culture medium was changed every two days with

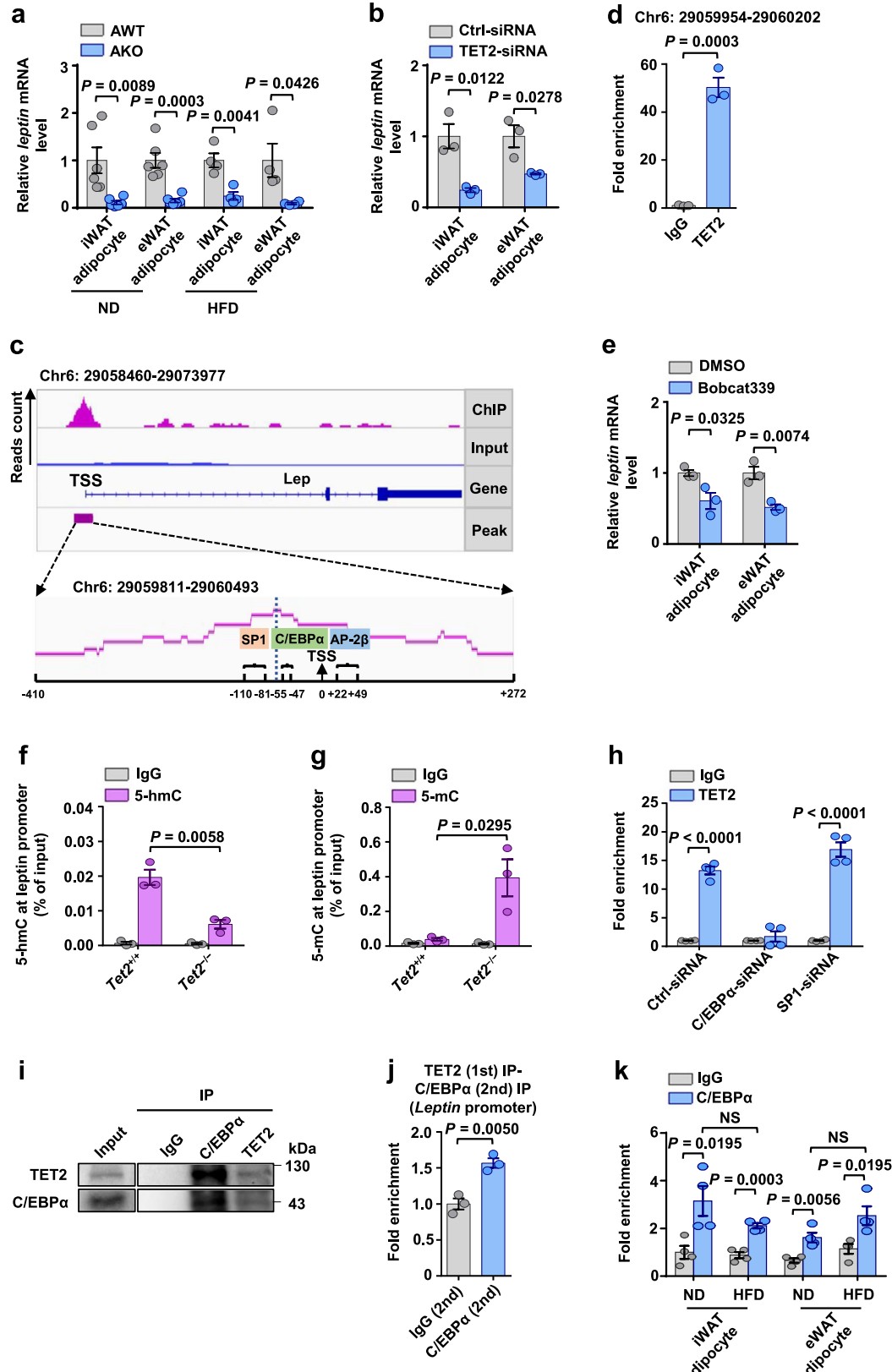

DMEM-High Glucose medium containing 10 µg/mL insulin for subsequent days.

## Adipose tissue fractionation

Adipose tissues were rinsed in PBS, minced, and digested with 1 mg/mL type II collagenase (Worthington, Cat#LS004177) and 1% bovine serum albumin for 30 min at 37 °C with gentle agitation (120 rpm). The digested tissue was passed through a 100 µm nylon mesh. After centrifugation at 300 g for 5 min, the upper floating mature adipocyte fraction was carefully removed and washed with PBS containing 2 mM EDTA. The lower SVF was collected and washed with PBS, subsequently incubated in red blood cell lysis buffer for 5 min on ice. The SVF was

**Fig. 6 | TET2 upregulates leptin gene expression via interacting with C/EBPα in adipocytes. a** *Leptin* mRNA levels relative to *36b4* in adipocytes of iWAT and eWAT from AWT and AKO mice fed either ND for 18 weeks or HFD for 12 weeks (*n* = 6 ND; *n* = 4 HFD). **b** *Leptin* mRNA levels in differentiated adipocytes from iWAT and eWAT treated with Ctrl-siRNA or TET2-siRNA for 24 h (*n* = 3). **c** TET2 ChIP-qPCR at the promoter of *leptin* gene in differentiated adipocytes (*n* = 3). **d** The binding domain between TET2 and leptin (TSS: Transcriptional start site). **e** mRNA levels of *leptin* relative to *36b4* in adipocytes from iWAT and eWAT treated with DMSO or Bobcat339 for 24 h (*n* = 3). **f** 5-hmC hMeDIP-qPCR at the promoter of *leptin* gene in mature adipocytes from *Tet2*[+/+] and *Tet2*[−/−] mice fed ND for 8 weeks (*n* = 3). **g** 5-mC hMeDIP-qPCR at the promoter of *leptin* gene in mature adipocytes from *Tet2*[+/+] and *Tet2*[−/−] mice fed ND for 8 weeks (*n* = 3). **h** TET2 ChIP-qPCR at the promoter of *leptin*

gene in differentiated adipocytes treated with Ctrl-siRNA, C/EBPα-siRNA or SP1-siRNA (*n* = 4). **i** C/EBPα-TET2 Co-IP. Endogenous immunoprecipitation assay with C/EBPα and TET2 in adipocytes from ND-fed mice. **j** ChIP-reChIP assay at *leptin* promoter. The ChIP-reChIP assay was performed in adipocytes from ND-fed mice. Anti-TET2 was used in the first immunoprecipitation, and anti-IgG and anti-C/EBPα were used in the second immunoprecipitation (*n* = 3). **k** C/EBPα ChIP-qPCR at the promoter of *leptin* gene in mature adipocytes of iWAT and eWAT from ND-fed and HFD-fed mice (*n* = 4). All data are presented as mean ± SEM. *n* indicates the number of biologically independent samples examined. *P*-values are indicated on the graph. Statistical values are determined by two-sided unpaired Student's *t*-test. Source data are provided as a Source Data File.

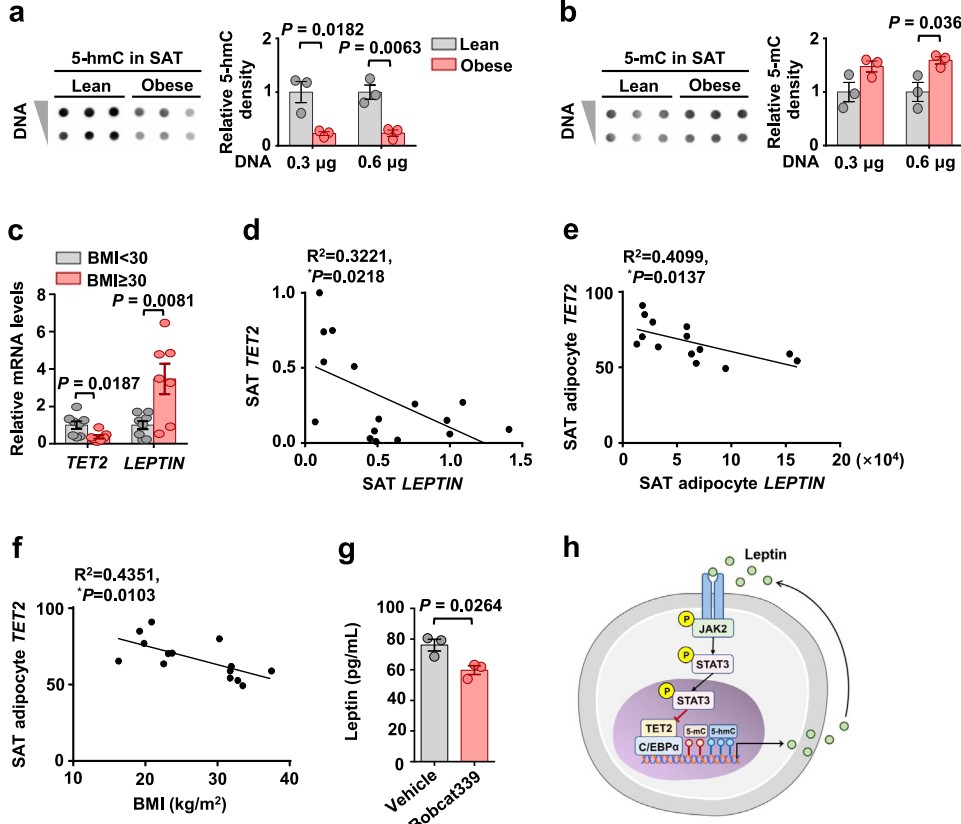

**Fig. 7 | TET2 levels are negatively correlated with LEPTIN levels and BMI in humans. a**, **b** Genomic 5-hmC (**a**) and 5-mC (**b**) levels in SAT from the lean and obese humans (*N* = 3). **c** mRNA levels of *TET2* and *LEPTIN* relative to *ACTB* in SAT (*N* = 8 nonobese and *N* = 7 obese subjects). **d** Correlation between *TET2* and *LEPTIN* in SAT. **e** Correlation between *TET2* and *LEPTIN* in adipocytes of SAT (Published microarray data: GSE44000). **f** *TET2* levels in human SAT adipocytes were negatively associated with BMI (Published microarray data: GSE44000). **g** Leptin

secretion in adipocytes from lean SAT treated with DMSO or Bobcat339 for 24 h (*N* = 3). **h** Model of the negative feedback loop between TET2 and leptin in adipocytes. All data are presented as mean ± SEM. *N* indicates the number of biologically independent samples examined. *P*-values are indicated on the graph. Statistical values are determined by two-sided unpaired Student's *t*-test in (**a**–**c**) and (**g**), two-sided Spearman's correlation analysis in (**d**–**f**). Source data are provided as a Source Data File.

then washed with PBS and collected by centrifugation at 500 g for 5 min. The resultant adipocytes and SVF were subjected to downstream analysis.

**Primary ASCs**

The SVF fraction was resuspended in DMEM/F12 plus 10% FBS, 100 IU/mL of penicillin, 100 mg/mL of streptomycin solution, and β-FGF (10 ng/mL, PeproTech, Cat#450-33,), at 5% $CO_2$ and 37 °C in a humidified cell incubator. After seeding, the medium was changed 24 h later to remove non-adherent cells and obtain mouse adipose-derived stem cells (ASCs). All experiments were conducted using cells at passage 3. For in vitro adipogenesis, confluent ASCs were used and treated with adipocyte differentiation induction cocktail: 0.5 mM IBMX, 1 μM dexamethasone, 170 nM insulin, followed by maintenance treatment

(10 μg/mL insulin) every 48 h. Once differentiated, differentiated adipocytes were treated with HFD-induced various factors or stained with Oil Red O. Differentiated adipocytes were serum deprived for 3 h with DMEM/F12 containing 0.2% FBS, and treated with 1 μg/mL leptin (Peprotech, Cat#45031), 5 ng/mL TNF-α (Peprotech, Cat#315-01 A), 500 ng/mL LPS (Sigma-Aldrich, Cat#L2630), 2 ng/mL IFN-γ (Peprotech, Cat#315-05) and 10 ng/mL TGF-β1 (Peprotech, Cat#100-21) for 24 h.

**In vivo adipogenesis**

ASCs from iWAT of *Tet2*[+/+] and *Tet2*[−/−] mice were purified by cell culture. The cells at passage 3 were identified as ASCs. 1 × 10[6] ASCs (donor cells) were resuspended in 100 μL Matrigel (356237, Corning) on ice. ASCs from *Tet2*[+/+] and *Tet2*[−/−] mice were injected into each flank of 3 week-old male C57BL/6J mice to compare the adipogenesis capacity in vivo

directly. After 5 weeks of HFD feeding, all Matrigel plugs were excised and fixed in 4% paraformaldehyde overnight, dehydrated, and embedded in paraffin. Sections of 5 μm were stained with hematoxylin and eosin.

### siRNA treatment
At day 8–10 of differentiation, to knockdown *Jak2*, *Stat3*, *C/ebpα, and Sp1* in differentiated adipocytes, Lipofectamine RNAiMAX (Thermo Fisher Scientific, Cat#13778075) was used. Lipofectamine RNAiMAX reagents and small interfering RNA (siRNA, RiboBio) were mixed in Opti-MEM (Gibco, Cat#31985070), and then added into culture medium. siRNA primer sequences are shown in Table S2.

### Neutralizing leptin antibody treatment
Neutralizing leptin antibody treatment was conducted using eWAT adipocytes from 8 week-old C57BL/6J mice. Mature adipocytes were treated with 0.2 μg/mL neutralizing leptin antibody (R&D, Cat#AF498) and IgG control antibody (BioXCell, Cat#BE0083) for 1 h, respectively, and then added CM from eWAT for another 24 h. For CM preparation, iWAT and eWAT from age-matched ND-fed and 12 week HFD-fed mice were minced and cultured with FBS-free DMEM for 24 h. CM from WATs was normalized by fat mass (10 mL of DMEM per 1 g of WATs)[17].

### Bobcat339 treatment
Bobcat339 treatment was conducted using the mature adipocytes from WATs of mice and SAT of healthy donors. 50–100 μL mature adipocytes were treated with 10 μM Bobcat339 (Topscience, Cat#T5198) and DMSO (Sigma-Aldrich, Cat#D2650) as a control in a 96-well plate for 24 h, respectively.

### Genomic DNA extraction and dot blot
Genomic DNA was extracted from WAT or adipocytes using a QIAamp DNA Mini kit (QIAGEN, Cat#51304). Take 0.1 μg, 0.3 μg or 0.6 μg of DNA respectively, add a certain volume of deionized water to make the volume of each sample 50 μL. 50 μL of DNA denaturation buffer (1 M NaOH, 25 mM EDTA) was added to the samples, and samples were then incubated at 100 °C for 10 min. Transfer the samples immediately to ice and add 50 μL of cold 2 M ammonium acetate (pH 7.0) for additional 20 min. Meanwhile, the nitrocellulose membrane was pre-soaked in double-distilled $H_2O$ and then in 6 × SSC buffer (0.9 M NaCl, 90 mM $Na_3C_6H_5O_7$) for 20 min. After setting up the dot blot apparatus, the membrane was rehydrated with 200 μL TE buffer. Then, the denatured DNA was applied and washed by 200 μL 2 × SSC buffer. Let the membrane dry and put the membrane between blotting papers which presoaked by 2 × SSC buffer for 20 min. Baking the membrane in the drier at 80 °C for 2 h. The membrane was blocked in 5% BSA in PBS and 0.1% Tween-20 for 1 h at room temperature. Apply the mouse 5-mC (Active Motif, Cat#61255) or 5-hmC (Active Motif, Cat#39791) primary antibody diluted 1:1000 in blocking buffer and incubated overnight at 4 °C on a shaker. Wash the membrane with TBST 3 times for 15 min each. Incubate with anti-HRP secondary antibody (Boster, Cat#BA1054) diluted 1:5000 in 2% BSA for 1 h at room temperature on a shaker. Wash the membrane as previously. Apply Clarity™ Western ECL Substrate (Bio-rad, Cat#170-5061) and expose to film.

### RNA isolation and qRT-PCR
Total RNA of cells and tissues was extracted by Trizol (Life Technologies, Cat#15596026) according to the manufacturer's protocol, cDNA was synthesized using a RevertAid First Strand cDNA Synthesis Kit (Thermo scientific, Cat#K1622). Real-Time PCR was performed using SYBR Green Master Mix (Vazyme, Cat#Q511-02) on Applied Biosystems ViiA™ 7 Real-Time PCR System. Normalized mRNA expression for mouse samples was calculated using *β-actin* or *36b4* as the reference gene. *β-ACTIN* were used as housekeeping genes for human samples. Relative mRNA expression was calculated using the ΔΔCt method. All

primers were designed by Sangon Biotech, and the sequence of primers were listed in Table S3.

### Gene expression quantification of Tets
We quantified the gene expression levels of *Tets* in individual samples obtained from both iWAT and eWAT of 7 week-old mice using the published bulk RNA-seq data (GSE132706). The SRA files of the GSE132706 dataset were retrieved from a public database utilizing the sra-tools software. These files were subsequently transformed into fastq format for upstream analysis. Alignment procedures were conducted using hisat2 2.2.1 software against the mm10 version of the reference genome. Quantification of gene expression from RNA-seq data was performed using featureCounts 2.0.6 to generate a count matrix file, employing the mm10 version of the gene annotation file. The TPM (Transcripts Per Million) values were computed using R 4.2.0 software[60].

### Western blot
Cells and tissues were lysed in RIPA buffer (Beijing Dingguo Changsheng Biotech, Cat#WB-0072) with protease inhibitor PMSF (Beyotime, Cat#ST505) and cOmplete™ EDTA-free (Sigma-Aldrich, Cat#4693132001). The protein concentration was determined with Pierce Microplate BCA Protein Assay kit (Thermo scientific, Cat#23252). 30 μg of total protein were separated in 8% SDS-polyacrylamide gel electrophoresis and transferred to PVDF membrane (Merck Millipore, Burlington, MA). The membranes were blocked in 5% BSA in PBS for 1 h at room temperature and then incubated overnight at 4 °C with the following primary antibodies: TET2 (Proteintech, Cat#21207-1-AP, 1:1000), C/EBPα (Cell Signaling Technology, Cat#8178, 1:1000), *β*-actin (Sigma-Aldrich, Cat#A5316, 1:40000). Wash the membrane with TBST 3 times for 15 min each. Incubate with anti-HRP secondary antibody (Boster, Cat#BA1054) diluted 1:5000 in 2% BSA for 1 h at room temperature on a shaker. Wash the membrane as previously. Apply Clarity™ Western ECL Substrate (Bio-rad, Cat#170-5061) and expose to film. Bio-Rad Image Lab and Image-Pro Plus software were used for analysis.

### H&E and immunohistochemical staining
Paraffin-embedded tissue blocks from mice were sectioned at 5 μm thickness, deparaffinized, rehydrated, and stained with hematoxylin and eosin. For TET2 immunohistochemistry, adipose tissue sections were heated at 60 °C for 1 h and deparaffinized using xylene and a series of gradient alcohols. Sections were blocked with 0.3% $H_2O_2$ (Boster, Cat#AR1108) for 20 min. To minimize non-specific staining, sections were blocked with 2% BSA in PBS for 30 min at room temperature. Incubation with the primary antibody (1:200 TET2, Abcam, Cat#94580) was performed overnight at 4 °C, followed by incubation with a secondary anti-rabbit antibody (Boster, Cat#BA1054, 1:200) for 1 h at room temperature. Finally, the slides were stained with the DAB peroxidase substrate kit (ZSGB bio, Cat#ZL2-9018) for 1 min. Images were viewed using an Olympus microscope and image acquisition was performed using Olympus microsystems imaging software. Data analysis was performed using the Image-Pro Plus 6.0.

For p-STAT3 immunohistochemistry, overnight fasted mice were injected with 2.5 mg/kg leptin. After 45 min, the mice were heart-perfused with physiological saline mixed with heparin, followed by paraformaldehyde. Then the brains were removed and sliced at 5 μm per slide to perform p-STAT3 (Cell Signaling Technology, Cat#9145, 1:200) staining.

### Metabolic cage studies
Metabolic analysis was initiated at a point when body weight differences were not yet observed in mice. All metabolic cage experiments in this study were conducted with a Comprehensive Lab Animal Monitoring System (CLAMS, Columbus Instruments, USA). Mice were fed

ad libitum and housed in the metabolic cages for 3 days. During this time, locomotor activity, $O_2$ consumption, and $CO_2$ consumption were measured. Energy expenditure was calculated based on the measurements of $O_2$ and $CO_2$.

## Body weight, body composition, food intake

Mouse body weight was monitored weekly. Mouse body composition was assessed by using the Minispec Body Composition Analyzer LF50 (Bruker, Germany). To measure food intake, the mice were individually housed. Before experiment, the mice were acclimated to the single cages for at least 3 days to minimize anxiety effects. Food intake and body weight were measured daily. For leptin-induced food intake experiments, the mice were fasted overnight. On the subsequent day, the mice were administered a dose of leptin at 1 µg/kg body weight, with a dose volume of 10 µL per gram of body weight. Food intake was measured at various time points as indicated in the figure.

## Glucose tolerance test (GTT) and insulin tolerance test (ITT)

For glucose tolerance tests, mice were fasted for 16 h, following they received intraperitoneal injection of 1 g/kg glucose. Blood samples taken from tail vein were measured at 0, 15, 30, 45, 60, and 120 min using a handheld glucometer (ACCU-CHEK active glucometer, Roche). For insulin tolerance tests, mice were fasted for 6 h and injected with 0.75 U/kg insulin. Blood samples taken from tail vein were measured at 0, 15, 30, 45, 60, and 90 min using Roche glucometer.

## Cold exposure studies

Male $Tet2^{-/-}$ mice and their control littermates (6 week-old), fed HFD for 5 weeks, were transferred from standard housing conditions to a 4 °C environment. They were individually housed in a non-bedded cage with free access to food and water. Male $Tet2^{-/-}$ ob/ob mice and $Tet2^{+/+}$ ob/ob mice (12 week-old) were singly housed without access to food or bedding, but with free access to water. At the end of the experiment, mice were sacrificed, and fat tissues (BAT, iWAT and eWAT) were isolated for gene expression analyses.

## Administration of leptin by intraperitoneal injection

The supplementation of leptin was initiated at a point when body weight differences were not yet observed in mice (5th week of HFD). The administered leptin used in this study was mouse pegylated leptin (Protein Laboratories Rehovot Ltd.), which prolongs the half-life of leptin. The specific protocol entailed daily intraperitoneal injections as follows: a dosage of 500 µg/kg body weight for the first two days, a dosage of 250 µg/kg body weight for the subsequent two days, followed by a daily dosage of 125 µg/kg body weight.

## ELISA assays

Plasma levels of mouse insulin (AiFang biological company, Cat#AF2579-A), plasma and CM levels of mouse leptin (Boster, Cat#EK0438) and CM levels of human leptin (4 A Biotech, Cat#CHE0053) were determined by ELISA using commercial kits.

## ChIP-seq analysis

For ChIP-seq, we employed a cell quantity of $5 \times 10^7$ primary ASCs that were induced to differentiate into mature adipocytes in vitro. ChIP was conducted using the EZ-Magna ChIP A/G kit (Cat#17-10086) from Millipore, USA. Following the procedure, the resulting DNA products were subjected to sequencing by the Beijing Boaojingdian Biotechnology Co., Ltd. Raw sequence quality was assessed with Fastqc software. NGS QC Toolkit[61] was used to remove poor quality sequences, Illumina-specific sequences and adapters from the reads. Reads were aligned against reference genome GRCm39 with bowtie1[62]. Peak calling and read density in peak regions were performed by MACS2[63]. R package ChIPseeker[64] was used for downstream analysis of peaks, such as annotations, visualizations binding site distribution relative to

features and obtaining enriched pathways. Transcription factors screening used bioinformatic portals, NCBI Gene database (https://www.ncbi.nlm.nih.gov/gene). TET2 peak in the proximal promoter region of the *leptin* gene was sequence region from 29060220 to 29073875. In this region, there are three published protein bind regions in the GenBank database, including SP1[65], C/EBPα[66], and AP-2β[45].

## ChIP−qPCR

For ChIP-qPCR, we employed a cell quantity of $1 \times 10^7$ primary ASCs that were induced to differentiate into mature adipocytes in vitro. For ChIP-reChIP[17], 400 µL mature adipocytes were used. The resulting DNA products were utilized for qPCR analysis. Antibodies against TET2 (Proteintech, Cat#21207-1-AP, 1:100), C/EBPa (GeenTex, GTX100674, 1:100), and normal rabbit IgG were used for immunoprecipitation. The primer sequences for qPCR were as follows: Forward: 5′-CTAGAATG GAGCACTAGGTTGC-3′, Reverse: 5′-CCCTCTTATAACTGCCCCAG-3′.

## Co-Immunoprecipitation

The Co-IP experiment was conducted using eWAT adipocytes from 8 week-old C57BL/6J mice. The Pierce Crosslink Magnetic IP/Co-IP Kit (Cat#88805) from Thermo Scientific, USA, was employed for this study. Immunoprecipitation was performed using 10 µg of rabbit anti-mouse TET2 antibody, C/EBPα antibody or rabbit IgG antibody. The obtained immunoprecipitates, along with the input protein samples, were subjected to Western blot analysis. The following antibodies were used for Co-IP: anti-TET2 (Proteintech, Cat#21207-1-AP) and anti-C/EBPα (GeneTex, Cat# GTX100674).

## hMeDIP-qPCR and MeDIP-qPCR

Genomic DNA was sheared by sonication to an average of 200−800 bp size. One microgram of denatured DNA was used for hMeDIP-qPCR MeDIP-qPCR. The hMeDIP assay was conducted using the hMeDIP kit from Active Motif (Cat#55010), while the MeDIP assay employed the MeDIP kit from Active Motif (Cat#55009). Recovered DNA was used for qPCR analysis. Primers for hMeDIP-qPCR and MeDIP-qPCR studies were as follows: Forward: 5′-CCAGTCTTTCTAATAGCACCCC-3′, Reverse: 5′-GAATTCAACACCAAGCTGTCC-3′. All data are normalized to input.

## Statistical analyses

Statistical analyses were performed with GraphPad Prism 9 and data was performed using Microsoft Office Excel 2021 (v.16.56). Two-sided unpaired Student's $t$-tests for pairwise comparison and Pearson correlation for linear regression, were performed to calculate statistical significance. When comparing Oxygen Consumption, Energy Expenditure, and Activity between different groups with body weight as a covariate, one-way ANCOVA was performed. Significance was considered as $P < 0.05$. Actual $P$-values were reported. Mice were randomly assigned to treatment groups for in vivo studies. n values represent independent biological replicates for cell experiments or individual mice for in vivo experiments.

## Reporting summary

Further information on research design is available in the Nature Portfolio Reporting Summary linked to this article.

# Data availability

The bulk RNA-seq data of iWAT and eWAT used in this study are available in the GEO database under accession number GSE132706. Hyperlink: https://www.ncbi.nlm.nih.gov/geo/query/acc.cgi?acc=GSE132706. The microarray data of human adipocytes[67] used in this study are available in the GEO database under accession number GSE44000. Hyperlink: https://www.ncbi.nlm.nih.gov/geo/query/acc.cgi?acc=GSE44000. ChIP-seq data of differentiated adipocytes

generated in this study have been deposited in the GEO database under accession number GSE252186. Hyperlink: https://www.ncbi.nlm.nih.gov/geo/query/acc.cgi?acc=GSE252186. All other data are available in the article and its supplementary files or from the corresponding author upon request. Source data are provided with this paper.

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

## Acknowledgements

We thank Dr. Guoliang Xu (Shanghai Institutes for Biological Sciences, Chinese Academy of Sciences) for generously providing the *Tet2*$^{-/-}$ mice. We thank Dr. Zhenqi Liu (Department of Medicine, University of Virginia Health System) and Dr. Wenwen Zeng (School of Medicine, Tsinghua University) for their scientific advice in the research. We thank Dr. Christian Wolfrum (Institute of Food Nutrition and Health, Eidgenössische Technische Hochschule Zürich) and Dr. Hua Dong (Institute of Food Nutrition and Health, Eidgenössische Technische Hochschule Zürich) for valuable guidance regarding the in vivo adipogenesis protocol. We thank Dr. Win Topatana (Zhejiang University) for his assistance in revising this paper. This work was supported by the National Key R&D Program of China (2020YFA0803604, T.D.), the National Natural Science Foundation of China, Key Program (82130024, T.D.), the Natural Science Foundation of Hunan Province (2023JJ40809, Q.Z.), the Natural Science Foundation of Changsha (74432, Q.Z.).

## Author contributions

Q.Z. and J.F.S. wrote the manuscript, designed and performed the most of the experiments, analyzed the data; X.X.S., D.D.W., X.Y.L. and W.Y.H. performed the animal experiments; Y.Y.J., W.Q.M., W.F.E.A.N. and Y.M. performed the cell experiments; F.Q.W., H.Z., L.M.X. and Y.T. conceptualized experiments and performed the ChIP-seq analysis; Y.J.D. and W.L. provided and prepared human adipose tissues, analyzed results; Z.G.X. and H.J.W. contributed to discussion and manuscript editing; T.D. conceptualized the study, supervised the study; all authors discussed and edited the manuscript.

## Competing interests

The authors declare no competing interests.
