## [Peer Review File · Nature Communications]

A Negative Feedback Loop Between TET2 and Leptin in Adipocyte Regulates Body WeightREVIEWER COMMENTS

Reviewer #1 (Remarks to the Author):

In the manuscript, Dr. Zeng et al investigated the negative feedback loop between TET2 and leptin in adipocytes. The authors show that deletion of Tet2 in adipocytes results in an anti-obese and anti-diabetic phenotype. These effects are dependent on reducing circulating leptin levels, as removing leptin component completely blocks the beneficial effects. Furthermore, the authors have shown a nice correlation in human population. In addition, the authors have investigated the mechanism how Tet2 regulates leptin expression in human adipocytes and revealed Tet2 interacts with C/EBP α to increase the hydroxymethylcytosine levels of the leptin gene promoter. This is an exacting study and the results presented in this manuscript fully support the conclusion they made. The manuscript is well-written and the manuscript is easy to read. However, there are some concerns need to be addressed:

1. As shown in Figure 1E, obesity-induced decline in adipocyte Tet2 levels is much later in eWAT than in iWAT. How to explain this disparity between eWAT and iWAT?
2. In Figure S1, the WB bands of β -actin are too faint to quantify.
3. The present data indicates that both whole-body and adipocyte-specific Tet2 deficiency alleviate the fat expansion caused by HFD feeding. What are the effects of Tet2 deficiency on the adipocyte size under ND- and HFD-fed conditions?
4. Hepatic fat accumulation has been assessed through H&E staining, which may not be sufficient to confirm the presence of fatty liver. It is advisable to conduct Oil Red O staining to specifically examine fat accumulation in the liver.
5. The immunohistochemistry images of p-STAT3 in Figure 5C are indistinct. Would the authors provide higher-resolution images?
6. Some critical information regarding the human donors, such as age, gender, BMI, and biochemical data, is currently absent.
7. This study demonstrates that TET2 regulates the expression of the leptin gene by increasing the levels of hydroxymethylcytosine at the leptin promoter. Nevertheless, the decline of TET2 does not suppress the increase levels of leptin expression in adipocytes during obesity. The authors speculate that the reduction of adipocyte TET2 induced by hyperleptinemia appears to be a compensatory response aimed at suppressing the elevated

levels of leptin. But they did not provide any evidence. Authors can either perform new experiments to support this idea or discuss this shortcoming.

Reviewer #2 (Remarks to the Author):

Comments

In this manuscript, the authors revealed a negative feedback loop between TET2 and leptin in adipocytes in obesity. However, how Tet2 regulates the function of adipocyte and metabolism in mice needs to be further investigated. Especially, browning of iWAT and eWAT as well as the function of BAT should be taken into consideration to account for increasing energy consumption and none change of iWAT weight in Tet AKO mice.

Furthermore, the mechanism of TET2-mediated function in adipocyte and leptin expression as well as metabolism were not fully explored.

Major:

1. The authors did not follow the “Sex and Gender Equality in Research – SAGER - guidelines”. Only female individuals were included in this study, and none explanation and discussion were provided for this study design. Thus, the results of this study were limited.
2. Line 127-128: The authors found that the expression of Tet was not changed upon HFD feeding in iWAT and eWAT. However, it was known that immune cells account for over 30% of SVF cells in iWAT and eWAT. More importantly, the number of immune cells significantly increased upon HFD feeding. Thus, the author should carefully check and exclude the immune cells account for observed changes of 5-hmC.
3. Line 197-200: Why the eWAT weights in AKO were reduced rather than the iWAT weights? It was showed in Figure 1E, the expression of Tet2 in iWAT was reduced at 4 to 12 weeks of HFD, while the expression of Tet2 in eWAT only reduced at 12 weeks of HFD. It seems that the loss of Tet2 in iWAT does not affect the function of iWAT. Besides, it was strange that the Tet2 expression in eWAT only loss in the 12 weeks of HFD (as shown in Figure 1E), but the weight significantly decreased at 12 weeks of HFD. The author should give a full explanation for these opposite results.
4. In Figure 3H-3J and Figure 4H-4K. The author found increased energy consumption in Tet2 KO and AKO mice. Dose the authors check the browning of iWAT and eWAT as well as the function of BAT? Usually, browning of iWAT and eWAT or improving function of BAT may

results in increased energy consumption.

5. In Figure 6H. The author claimed that TET2 interacted with C/EBP α to regulate leptin expression. However, the mechanism was not clearly. Does the author check the interaction between C/EBP α and TET2? Did TET2 affect the expression of C/EBP α ? Did C/EBP α affect the expression of TET2?

6. Whether the enzyme activity of TET2 was required for the leptin expression and metabolism regulation? Does Mutant assay of TET2 should be performed to explain this question? Besides, the author does not identify the directly role of 5hmC in the function of adipocyte and metabolism? Further assay should be performed to prove the role of TET2-mediated 5hmC in adipocyte and metabolism.

Minor:

1. Figure S1D: What does the mRNA level (Relative counts of Tet genes) mean? Generally, TPM or FPKM was used for differential analysis.

2. The western-blot results of TET2 and actin in Figure 1D and Figure S1E and S1H were blurring. The author should replace the blot results.

3. Line 133-139: Why chose 1 μ g/ml Leptin for cell assay? What's the concentration of leptin in serum and adipose tissue? The author should consider the concentration of leptin in vivo, and design in vitro assay to test the function of leptin.

4. What does the conditioned medium (CM) from obese WAT mean?

5. It was hard to see the p-STAT3 staining in Figure 5C. The author should replace the results.

6. Line 257-258: It's unclear that how SP1, C/EBP α , and AP-2 β were screen out. More details should be given to explain the results.

7. The results of Figure 6H were blurring. The author should replace the blot results.

8.

9. Line 270-272. It was shown in Figure 4 that knockout of Tet2 did not affect the weight of iWAT which indicate that Tet2 did not play any function in iWAT. So, It may be meaningless to explore the expression of Tet2 in iWAT.

Reviewer #3 (Remarks to the Author):

Understanding leptin resistance in obesity is of critical importance. The current study proposes that TET2 plays an important role in mediating leptin resistance in obesity. The

main findings include:

1. TET2 and 5hmC levels in adipocytes gradually decrease in response to high-fat feeding, while they show an increased level in ob/ob mice.
2. Leptin suppresses Tet2 via the JAK2-STAT3 signaling pathway in adipocytes.
3. General and adipocyte-specific Tet2 deficiency attenuates diet-induced obesity and insulin resistance by promoting energy expenditure and inhibiting food intake.
4. Tet2 deficiency partially improves leptin resistance by reducing leptin levels.

The overall study design is thorough, and the presented data are highly significant and supportive. I address the following points to strengthen the authors' conclusions:

1. TET2 is expressed in many cell types and tissues. Extra-adipose expression could be an important factor in the metabolic outcomes in Tet2^{-/-} mice. For instance, studies have shown that TET2 regulates brain function, such as cognitive function, albeit with some contradictory findings. Were there differences in physical activity between the genotypes? TET2^{-/-} mice have been shown to aggravate insulin resistance and obesity by leading to clonal expansion of hematopoiesis. Due to such complexity, it would be more appropriate to investigate the metabolic phenotype in Adi-Tet2 KO x ob/ob mice, rather than using whole-body Tet2 mice.
2. What is the underlying basis for increased energy expenditure and reduced obesity by Tet2 deficiency and what is the tissue level of leptin sensitivity in various tissues including BAT and brain? TET2 is known to promote adipogenesis in vitro. Reduced adiposity due to Tet2 deficiency could also be attributed to decreased adipogenesis. Therefore, in vivo and/or ex vivo adipogenesis should be assessed to elucidate its contribution. The effect on adaptive thermogenesis should be characterized in Tet2-deficient and double mutant mice.
3. Leptin regulates fertility. Are there any impact on fertility by Tet2 deficiency?
4. In addition to the snapshot of TET2 binding at Lep locus, comprehensive data analysis

should be presented.

5. How does the binding of C/EBP α to Lep change in obesity compared to lean conditions?

ChIP-reChIP studies will be essential to investigate the co-recruitment of C/EBP α and TET2 at Lep.

REVIEWER COMMENTS

Reviewer #1 (Remarks to the Author):

In the manuscript, Dr. Zeng et al investigated the negative feedback loop between TET2 and leptin in adipocytes. The authors show that deletion of Tet2 in adipocytes results in an anti-obese and anti-diabetic phenotype. These effects are dependent on reducing circulating leptin levels, as removing leptin component completely blocks the beneficial effects. Furthermore, the authors have shown a nice correlation in human population. In addition, the authors have investigated the mechanism how Tet2 regulates leptin expression in human adipocytes and revealed Tet2 interacts with C/EBP α to increase the hydroxymethylcytosine levels of the leptin gene promoter. This is an exacting study and the results presented in this manuscript fully support the conclusion they made. The manuscript is well-written and the manuscript is easy to read. However, there are some concerns need to be addressed:

Thank you for your very helpful and thorough comments.

1. As shown in Figure 1E, obesity-induced decline in adipocyte Tet2 levels is much later in eWAT than in iWAT. How to explain this disparity between eWAT and iWAT?

Response:

To explain the temporal discrepancy in the reduction of *Tet2* levels in iWAT and eWAT adipocytes during obesity, we investigated the leptin levels in these two types of adipocytes during obesity. Our results revealed that the mRNA levels of *leptin* increased in both iWAT and eWAT adipocytes at 2 weeks of HFD and continued to rise until 12 weeks of HFD (Figures S2b and S2c). The secretion levels of leptin from these adipose tissues were also similar under obese conditions (Figure 2d). However, as shown in Figure S2A, leptin suppressed *Tet2* expression more prominently in iWAT adipocytes compared to eWAT adipocytes. This suggests that iWAT adipocytes are more sensitive to leptin-induced TET2 inhibition. Therefore, it appears that the reduced sensitivity to leptin, rather than leptin concentration, contributes to the delayed decline of adipocyte *Tet2* levels in eWAT during obesity.

2. In Figure S1, the WB bands of β -actin are too faint to quantify.

Response:

We have replaced the blot results with more discernible figures and re-evaluated the WB bands for quantification.

3. The present data indicates that both whole-body and adipocyte-specific Tet2 deficiency alleviate the fat expansion caused by HFD feeding. What are the effects of Tet2 deficiency on the adipocyte size under ND- and HFD-fed conditions?

Response:

As the reviewer suggested, we measured the diameters of adipocytes in *Tet2* deficient mice and control littermates under ND- and HFD-fed conditions. After HFD feeding, in whole-body *Tet2* deficient mice, the adipocyte size of iWAT and eWAT was smaller compared to WT mice. Additionally, in adipocyte-specific *Tet2* deficient mice, the adipocyte size of eWAT was smaller than that in control mice (Figures S4d and S6d). However, under ND-fed condition, the adipocyte size of iWAT and eWAT showed no difference between *Tet2* deficient mice and WT

mice (Figures S3c and S5f). These data are now added to the Results section.

4. Hepatic fat accumulation has been assessed through H&E staining, which may not be sufficient to confirm the presence of fatty liver. It is advisable to conduct Oil Red O staining to specifically examine fat accumulation in the liver.

Response:

We performed Oil Red O staining on livers from *Tet2* deficient mice and WT mice, and added these data to Figure S4e and S6e. These findings are consistent with the results from H&E staining.

5. The immunohistochemistry images of p-STAT3 in Figure 5C are indistinct. Would the authors provide higher-resolution images?

Response:

As the reviewer suggested, we have provided higher-resolution images in Figure 5c.

6. Some critical information regarding the human donors, such as age, gender, BMI, and biochemical data, is currently absent.

Response:

Information concerning clinical parameters of human volunteers have been included in Table S1.

Clinical parameters	Subject population (N=15)	
	Nonobese (N=8) Mean ± SE	Obese (N=7) Mean ± SE
Age (years)	40.8 ± 2.7	30.2 ± 3.9
Sex (Female/Male)	4/4	3/4
BMI (kg/m ²)	23.1 ± 1.3	37.2 ± 3.8*
TC (mmol/L)	3.7 ± 0.3	3.9 ± 0.2
TG (mmol/L)	1.0 ± 0.2	2.0 ± 0.5*
HDL (mmol/L)	1.1 ± 0.1	0.9 ± 0.1
LDL (mmol/L)	2.2 ± 0.2	2.6 ± 0.2*

Table S1. Phenotypic parameters of the nonobese (BMI <30 kg/m²) and obese (BMI ≥30 kg/m²) human subjects. Asterisks indicate significant differences ($P < 0.05$ by T-test) between nonobese and obese subjects. Phenotype abbreviations: body mass index (BMI), total cholesterol (TC), triglycerides (TG), high-density lipoprotein cholesterol (HDL) and low-density lipoprotein cholesterol (LDL). Data represent Means±SEM.

7. This study demonstrates that TET2 regulates the expression of the leptin gene by increasing the levels of hydroxymethylcytosine at the leptin promoter. Nevertheless, the decline of TET2 does not suppress the increase levels of leptin expression in adipocytes during obesity. The authors speculate that the reduction of adipocyte TET2 induced by hyperleptinemia appears to be a compensatory response aimed at suppressing the elevated levels of leptin. But they did not provide any evidence. Authors can either perform new experiments to support this idea or discuss this shortcoming.

Response:

This concern has been addressed in the final paragraph of the Discussion section, now

highlighted in red. The discussion is as follows: “In this negative feedback loop, leptin plays a predominant role, as leptin inhibits adipocyte TET2 significantly in the context of obesity. The hyperleptinemia-induced reduction of adipocyte TET2 appears to be a compensatory response aimed at suppressing the elevated levels of leptin. Although the decrease in TET2 expression in adipocytes did not prevent the increase in leptin levels during obesity, the loss of *Tet2* in obese mice resulted in the downregulation of leptin levels and an enhancement of leptin sensitivity, which ultimately led to weight loss. This suggests that the downregulation of TET2 in adipocytes can promote weight loss by reducing leptin levels.” Therefore, in obesity, the balance of this negative feedback loop is disrupted, making it challenging for the downregulation of TET2 to suppress the continuously rising leptin levels. Examination of the effects of TET2 overexpression in adipocytes on the levels of leptin during obesity would help address this point.

Reviewer #2 (Remarks to the Author):

Comments

In this manuscript, the authors revealed a negative feedback loop between TET2 and leptin in adipocytes in obesity. However, how *Tet2* regulates the function of adipocyte and metabolism in mice needs to be further investigated. Especially, browning of iWAT and eWAT as well as the function of BAT should be taken into consideration to account for increasing energy consumption and none change of iWAT weight in *Tet* AKO mice. Furthermore, the mechanism of TET2-mediated function in adipocyte and leptin expression as well as metabolism were not fully explored.

Thank you for your excellent review.

Major:

1. The authors did not follow the “Sex and Gender Equality in Research – SAGER - guidelines”. Only female individuals were included in this study, and none explanation and discussion were provided for this study design. Thus, the results of this study were limited.

Response:

Previous studies have demonstrated that mice fed a HFD exhibited a significantly higher percentage of visceral fat, body fat (Elise Jeffery, et al. *Cell Metab.* 2016 Jul 12;24(1):142-50.), weight gain (Abigail E. Salinero, et al. *Int J Obes (Lond).* 2018 Jun;42(5):1088-1091) and worsened insulin sensitivity (Medrikova, D., et al. *Int J Obes (Lond).* 2012 Feb 36(2): 262-272) in males compared to females. Therefore, we utilized male mice to investigate the role of TET2 in obesity. The discussion of this limitation has been added in the Methods section. However, our human data include both male and female volunteers, and both sexes show the same trend in the change of *TET2* and *LEPTIN* expression levels. Information about the gender of humans has been added to Supplementary Table 1.

2. Line 127-128: The authors found that the expression of *Tet* was not changed upon HFD feeding in iWAT and eWAT. However, it was known that immune cells account for over 30% of SVF cells in iWAT and eWAT. More importantly, the number of immune cells significantly increased upon HFD feeding. Thus, the author should carefully check and exclude the immune cells account for observed changes of 5-hmC.

Response:

As shown in Figures 1a and 1b, the levels of 5-hmC were significantly lower in purified adipocytes of iWAT and eWAT from obese mice compared to lean controls. This result could exclude the immune cells' effects in the changes of 5-hmC in WATs.

3. Line 197-200: Why the eWAT weights in AKO were reduced rather than the iWAT weights?

Response:

Our newly added results revealed significantly elevated mRNA levels of thermogenic genes in eWAT of AKO mice, a phenomenon not observed in iWAT (Figures 4m and 4n). This result suggests that leptin-induced thermogenesis primarily occurs in eWAT rather than iWAT in AKO mice, resulting in a more pronounced decrease in eWAT weights rather than the iWAT weights.

It was showed in Figure 1E, the expression of *Tet2* in iWAT was reduced at 4 to 12 weeks of HFD, while the expression of *Tet2* in eWAT only reduced at 12 weeks of HFD. It seems that the loss of *Tet2* in iWAT does not affect the function of iWAT. Besides, it was strange that the *Tet2* expression in eWAT only loss in the 12 weeks of HFD (as shown in Figure 1E), but the weight significantly decreased at 12 weeks of HFD. The author should give a full explanation for these opposite results.

Response:

As shown in Figure 6a, under both lean and obese conditions, AKO mice exhibited reduced *leptin* mRNA levels in iWAT adipocytes compared to AWT mice. This result confirms that the loss of *Tet2* in iWAT leads to the suppression of leptin expression, which contributes to the reduction of circulating leptin levels in *Tet2* deficient mice. In the negative feedback loop between leptin and TET2, leptin plays a predominant role, as leptin inhibits adipocyte TET2 significantly in the context of obesity. The hyperleptinemia-induced reduction of adipocyte TET2 appears to be a compensatory response aimed at suppressing the elevated levels of leptin. Although the decrease in TET2 expression in adipocytes did not prevent the increase in leptin levels during obesity, the loss of *Tet2* in obese mice resulted in the downregulation of leptin levels and an enhancement of leptin sensitivity, which ultimately led to weight loss. Additionally, compared to HFD-induced decline of adipocyte TET2, specific knockout of *Tet2* in adipocytes causes an earlier decrease and more pronounced decline in *Tet2* expression. Therefore, obese AKO mice exhibited lower circulating leptin levels and enhanced leptin sensitivity, which contributed to leptin-regulated thermogenic metabolism in eWAT, subsequently leading to reduced eWAT weights.

4. In Figure 3H-3J and Figure 4H-4K. The author found increased energy consumption in *Tet2* KO and AKO mice. Dose the authors check the browning of iWAT and eWAT as well as the function of BAT? Usually, browning of iWAT and eWAT or improving function of BAT may results in increased energy consumption.

Response:

Thank you for your thoughtful suggestions. We checked the expression levels of thermogenic genes in BAT, iWAT and eWAT from *Tet2* KO, AKO mice and their control mice after 5 weeks of HFD feeding. HFD-fed *Tet2*^{-/-} mice expressed higher mRNA levels of thermogenic genes in BAT (e.g., *Ucp1* and *Cidea*), iWAT (e.g., *Prdm16*, *Ppargc1a* and *Cidea*) and eWAT (e.g.,

Ucp1, *Prdm16* and *Pparg1a*) compared to *Tet2*^{+/+} mice (Figures 3l-3n). HFD-fed AKO mice displayed higher mRNA levels of thermogenic genes, such as *Prdm16*, *Pparg1a* and *Cidea* in BAT and *Ucp1*, *Pparg1a* and *Dio2* in eWAT, when compared to WT controls (Figures 4l-4n). These results suggest that *deletion* of *Tet2* leads to an increase in BAT activity and WAT browning during obesity, consequently resulting in increased energy expenditure and alleviated obesity. These data are now added to the Results section.

5. In Figure 6H. The author claimed that TET2 interacted with C/EBP α to regulate leptin expression. However, the mechanism was not clearly. Does the author check the interaction between C/EBP α and TET2?

Response:

The Co-IP experiment validated an interaction between TET2 and C/EBP α in mature adipocytes (Figure 6i). To corroborate the co-recruitment of C/EBP α and TET2 at the *leptin* gene promoter, we conducted a ChIP-reChIP experiment, reconfirming this interaction (Figure 6j). Combining these results with data from Figures 6f and 6g, our findings substantiate that TET2 directly interacts with C/EBP α to increase hydroxymethylcytosine levels and *leptin* mRNA expression by targeting the *leptin* gene promoter.

Did TET2 affect the expression of C/EBP α ? Did C/EBP α affect the expression of TET2?

Knockdown of *Tet2* gene didn't influence the expression of C/EBP α in adipocytes (Figure S9d). Similarly, knockdown of *Cebpa* gene didn't impact the expression of TET2 in adipocytes (Figure S9e). These findings suggest that TET2 and C/EBP α don't affect each other's expression levels in adipocytes. These data are now added to the Results section.

6. Whether the enzyme activity of TET2 was required for the leptin expression and metabolism regulation? Does Mutant assay of TET2 should be performed to explain this question? Besides, the author does not identify the directly role of 5hmC in the function of adipocyte and metabolism? Further assay should be performed to prove the role of TET2-mediated 5hmC in adipocyte and metabolism.

Response:

Thank you very much for your insightful suggestions. We can test whether the enzyme activity of TET2 is required for leptin expression and metabolism regulation through mutant assay of TET2 or treatment of TET2 inhibitor. Given the difficulty of transfecting mature adipocytes with plasmids, we chose an alternative approach. We used a TET2 enzymatic inhibitor, Bobcat339 (Gabriella N L Chua, et al. ACS Med Chem Lett. 2019 Jan 31;10(2):180-185), to assess its impact on *leptin* expression and 5-hmC levels in adipocytes. Treatment with Bobcat339 reduced levels of 5-hmC (Figure S9c) and decreased mRNA levels of *leptin* (Figure 6e) in mature adipocytes from both iWAT and eWAT. This suggests that TET2-mediated 5-hmC modification plays a regulatory role in leptin expression. These data are now added to the Results section.

Minor:

1. Figure S1D: What does the mRNA level (Relative counts of Tet genes) mean? Generally, TPM or FPKM was used for differential analysis.

Response:

Thank you for your valuable suggestions. Following your suggestion, we used TPM to quantify the gene expression levels of *Tets* in individual samples obtained from both iWAT and eWAT of 7-week-old mice. This analysis was conducted using the publicly available bulk RNA-seq data (GSE132706). Additional details on the data analysis procedures have been incorporated into the Methods section.

2. The western-blot results of TET2 and actin in Figure 1D and Figure S1E and S1H were blurring. The author should replace the blot results.

Response:

We have replaced the blot results with more discernible figures and re-evaluated the WB bands for quantification.

3. Line 133-139: Why chose 1 μ g/ml Leptin for cell assay? What's the concentration of leptin in serum and adipose tissue? The author should consider the concentration of leptin in vivo, and design in vitro assay to test the function of leptin.

Response:

Based on our previous experience (Tuo Deng, et al. Cell Metab. 2013 Mar 5;17(3):411-22.), we chose a concentration of 1 μ g/mL leptin for the cell assay. Actually, we conducted a dose-dependent assay and identified the effective starting concentration of leptin to be 120 ng/mL (Figure 2c). Following your suggestion, we have detected the plasma leptin levels in obese mice and found it to be lower at 20 ng/mL (Figure 5a). This concentration is insufficient to exert the inhibitory effect on TET2 expression. We measured the concentration of leptin in iWAT and eWAT from obese mice, which both approximately 300 ng/mL (Figure 2d), a level adequate to induce the inhibition on TET2 expression.

4. What does the conditioned medium (CM) from obese WAT mean?

Response:

We apologize for the confusion. The conditioned medium (CM) from obese WAT means the CM collected from iWAT and eWAT of HFD-fed mice. More details on CM preparation procedures have been added to the section of Methods: For CM preparation, iWAT and eWAT from age-matched ND-fed and 12-week HFD-fed mice were minced and cultured with DMEM for 24 h. CM from WATs was normalized by fat mass (10 mL of DMEM per 1 g of WATs).

5. It was hard to see the p-STAT3 staining in Figure 5C. The author should replace the results.

Response:

As the reviewer suggested, we have provided higher-resolution images in Figure 5c.

6. Line 257-258: It's unclear that how SP1, C/EBP α , and AP-2 β were screen out. More details should be given to explain the results.

Response:

Thank you for your invaluable suggestions. More details on how SP1, C/EBP α , and AP-2 β were screen out have been added to the section of Methods: Transcription factors screening used bioinformatic portals, NCBI Gene database (<https://www.ncbi.nlm.nih.gov/gene>). TET2

peak in the proximal promoter region of the leptin gene was sequence region from 29060220 to 29073875. In this region, there are three published protein bind regions in the GenBank database, including SP1, C/EBP α , and AP-2 β .

7. The results of Figure 6H were blurring. The author should replace the blot results.

Response:

We have replaced the blot results with more discernible figures in 6i.

8. Line 270-272. It was shown in Figure 4 that knockout of Tet2 did not affect the weight of iWAT which indicate that Tet2 did not play any function in iWAT. So, It may be meaningless to explore the expression of Tet2 in iWAT.

Response:

As shown in Figure 6a, under both lean and obese conditions, AKO mice exhibited reduced *leptin* mRNA levels in iWAT adipocytes compared to AWT mice. This result confirms that the loss of *Tet2* in iWAT leads to the suppression of leptin expression, which contributes to the reduction of circulating leptin levels in *Tet2* deficient mice.

Reviewer #3 (Remarks to the Author):

Understanding leptin resistance in obesity is of critical importance. The current study proposes that TET2 plays an important role in mediating leptin resistance in obesity. The main findings include:

1. TET2 and 5hmC levels in adipocytes gradually decrease in response to high-fat feeding, while they show an increased level in ob/ob mice.
2. Leptin suppresses Tet2 via the JAK2-STAT3 signaling pathway in adipocytes.
3. General and adipocyte-specific Tet2 deficiency attenuates diet-induced obesity and insulin resistance by promoting energy expenditure and inhibiting food intake.
4. Tet2 deficiency partially improves leptin resistance by reducing leptin levels.

The overall study design is thorough, and the presented data are highly significant and supportive. I address the following points to strengthen the authors' conclusions:

Thank you for your positive comments and invaluable suggestions. We believe the manuscript is now much improved.

1. TET2 is expressed in many cell types and tissues. Extra-adipose expression could be an important factor in the metabolic outcomes in *Tet2*^{-/-} mice. For instance, studies have shown that TET2 regulates brain function, such as cognitive function, albeit with some contradictory findings. Were there differences in physical activity between the genotypes?

Response:

We totally agree with your thoughts. We have analyzed the data from metabolic cage study. There is no significant difference in physical activity between *Tet2*^{+/+} and *Tet2*^{-/-} mice (Figure S4f). This data is now added to the Results section.

TET2^{-/-} mice have been shown to aggravate insulin resistance and obesity by leading to clonal expansion of hematopoiesis. Due to such complexity, it would be more appropriate to

investigate the metabolic phenotype in Adi-Tet2 KO x ob/ob mice, rather than using whole-body Tet2 mice.

Response:

We totally agree that it would be more appropriate to investigate the metabolic phenotype in Adi-Tet2 KO x ob/ob mice. Given the long mating duration required to obtain Adi-Tet2 KO x ob/ob mice and the extremely low production of Adi-Tet2 KO x ob/ob pups in each generation, we used whole-body *Tet2* deficient x ob/ob mice instead of Adi-Tet2 KO x ob/ob mice to define the role of leptin in *Tet2* deficient x ob/ob mice. However, in order to address this weakness, we supplemented HFD-fed AKO mice with leptin by long-term intraperitoneal injection to reach plasma leptin levels similar to PBS-injected AWT mice. Therefore, these results suggest that either leptin deficiency or leptin supplement normalized the metabolic changes induced by *Tet2* deficiency.

2. What is the underlying basis for increased energy expenditure and reduced obesity by *Tet2* deficiency and what is the tissue level of leptin sensitivity in various tissues including BAT and brain?

Response:

Thank you for your constructive suggestions. Our newly added results suggest that deletion of *Tet2* in adipocytes lead to the browning of iWAT and eWAT during obesity (Figures 3l-3n and 4l-4n), consequently resulting in increased energy expenditure.

Following your suggestion, we assessed leptin sensitivity in BAT and found no differences in P-STAT3 levels in BAT between *Tet2* deficient mice and control mice. This data was attached as below. Previous studies showed that leptin-regulated thermogenic metabolism in BAT and WATs is mediated by leptin-induced sympathetic nerve activation (Putianqi Wang, et al. Nature. 2020 Jul;583(7818):839-844). This sympathoexcitatory actions of leptin result from the stimulation of leptin receptor in the central nervous system. Leptin receptor is expressed at high levels in hypothalamus. Thus, the improvement of leptin sensitivity caused by *Tet2* deficiency during obesity occurs in hypothalamus rather than BAT.

Representative immunoblot images and densitometry analysis of P-STAT3 in BAT from AWT and AKO mice injected with PBS or leptin (n = 3 mice/group).

TET2 is known to promote adipogenesis in vitro. Reduced adiposity due to *Tet2* deficiency could also be attributed to decreased adipogenesis. Therefore, in vivo and/or ex vivo adipogenesis should be assessed to elucidate its contribution.

Response:

Thank you very much for your insightful suggestions. We collected adipose stem cells (ASCs) from whole body *Tet2*-deficient mice to conduct *in vivo* and *ex vivo* adipogenesis experiments, confirming that *Tet2* deficiency effectively reduced adipogenesis (Figures S4g-S4j). These results suggest that adipogenesis may contribute to the reduced obesity in whole body *Tet2*-deficient mice. However, treatment of ASCs from adipocyte-specific *Tet2* deficient mice with adipocyte differentiation induction cocktail resulted in no difference in adipogenesis compared with ASCs from AWT mice (Figures S6f and S6g). These outcomes indicate that the reduced adiposity in AKO mice is not attributed to adipogenesis. These data are now added to the Results section.

The effect on adaptive thermogenesis should be characterized in *Tet2*-deficient and double mutant mice.

Thank you very much for your construction suggestions. We checked adaptive thermogenesis in *Tet2*-deficient and double mutant mice. The cold-induced thermogenesis response in *Tet2* deficient mice was enhanced than that of the control groups after 5 weeks of HFD feeding (Figures S7g-S7j). However, no such difference was observed between double mutant mice and their control counterparts (Figures S7k-7n). This suggests that the enhanced thermogenic capacity resulting from *Tet2* deficiency is mediated by leptin. These data are now added to the Results section.

3. Leptin regulates fertility. Are there any impact on fertility by *Tet2* deficiency?

Response:

We used heterozygous *Tet2* knockout mouse (*Tet2*^{+/-}) mice to breed and obtain *Tet2*^{+/+} and *Tet2*^{-/-} mice. We generated adipocyte-specific *Tet2* knockout mice (AKO) by crossing *Tet2*^{fl/fl} mice to *Adipoq-Cre* transgenic mice. The deletion of *Tet2* does not impact the fertility of mice, including the numbers of pups and morphological differences. The details of the pups are as follows:

Male	Female	AWT	AKO
9	13	13	9
16	11	9	18
16	11	15	12
6	5	5	6
15	11	16	10
18	12	17	13
24	14	16	22
13	12	12	13
16	17	15	18
19	23	24	18
12	20	18	14
8	4	7	5
15	23	19	19
6	7	8	5
P value of Male vs Female: 0.7375		P value of AWT vs AKO: 0.6767	

4. In addition to the snapshot of TET2 binding at Lep locus, comprehensive data analysis should be presented.

Response:

We appreciate the valuable suggestions. In Figures S9a and S9b, we have presented comprehensive data of ChIP-seq. ChIP-seq signal in a 6 kb region flanking significant TET2 peak near the transcriptional start site (Figure S9a). Gene ontology analysis revealed that TET2 antibody immunoprecipitates were enriched in genes associated with cellular process, cell communication and structure development (Figure S9b). These data are now added to the Results and Methods sections.

5. How does the binding of C/EBP α to Lep change in obesity compared to lean conditions? ChIP-reChIP studies will be essential to investigate the co-recruitment of C/EBP α and TET2 at Lep.

Response:

We performed ChIP-qPCR to assess the enrichment of C/EBP α at the promoter of the *leptin* gene in adipocytes of WATs from both ND-fed and HFD-fed mice. As illustrated in Figure 6k, the binding capacity of C/EBP α to Lep showed no difference between obesity and lean conditions. Following your suggestion, we conducted a ChIP-reChIP experiment, confirming the co-recruitment of C/EBP α and TET2 at the *leptin* gene promoter (Figure 6j). More details are available in the Results and Methods sections.

REVIEWERS' COMMENTS

Reviewer #1 (Remarks to the Author):

The authors have successfully addressed all the concerns I have. I have no further concern raised.

Reviewer #2 (Remarks to the Author):

The authors have addressed my concerns. I have no other comments. Thanks!

Reviewer #3 (Remarks to the Author):

Most of the comments were extensively addressed; however, some essential information regarding ChIP-seq analysis is still missing. Specifically, details about the number of detected TET2 peaks, their genomic distribution (e.g., at enhancers vs. promoters), and the associated genes are needed. It would be informative to present representative peaks as well.

REVIEWER COMMENTS

Reviewer #1 (Remarks to the Author):

The authors have successfully addressed all the concerns I have. I have no further concern raised.

Response:

Thank you for your constructive comments and kind evaluation.

Reviewer #2 (Remarks to the Author):

The authors have addressed my concerns. I have no other comments. Thanks!

Response:

We greatly appreciate your kind comments.

Reviewer #3 (Remarks to the Author):

Most of the comments were extensively addressed; however, some essential information regarding ChIP-seq analysis is still missing. Specifically, details about the number of detected TET2 peaks, their genomic distribution (e.g., at enhancers vs. promoters), and the associated genes are needed. It would be informative to present representative peaks as well.

Response:

We greatly appreciate your valuable comments. The essential information regarding ChIP-seq analysis, such as details about the number of detected TET2 peaks, TET2 peaks over chromosomes, their genomic distribution (e.g., at enhancers vs. promoters), and the associated genes, has now been added to the Results, Figures (Figure S9b and S9c) and Supplementary Table (Table S4).